# Occurrence of Moulds and Yeasts in the Slaughterhouse: The Underestimated Role of Fungi in Meat Safety and Occupational Health

**DOI:** 10.3390/foods14081320

**Published:** 2025-04-11

**Authors:** Melissa Alves Rodrigues, Pedro Teiga-Teixeira, Alexandra Esteves

**Affiliations:** 1Netherlands Food and Consumer Product Safety Authority (NVWA), P.O. Box 43006, 3540 AA Utrecht, The Netherlands; 2Department of Veterinary Sciences, University of Trás-os-Montes and Alto Douro, 5001-801 Vila Real, Portugal; alexe@utad.pt; 3Animal and Veterinary Science Centre (CECAV), Associate Laboratory for Animal and Veterinary Sciences (AL4AnimalS), University of Trás-os-Montes and Alto Douro, 5001-801 Vila Real, Portugal

**Keywords:** mould, yeast, slaughterhouse, meat, occupational, food safety, public health

## Abstract

Despite their potential impact on meat safety and occupational health, fungi are often underestimated contaminants in slaughterhouses. Moulds and yeasts may be associated with meat contamination in multiple processing stages, and mycotoxigenic species, such as *Aspergillus*, *Fusarium*, and *Penicillium*, pose food safety concerns. Bioaerosols may carry infectious fungi at the slaughterhouse that are capable of causing respiratory conditions and allergies. Chronic exposure to mycotoxins can have hepatotoxic, nephrotoxic, and carcinogenic effects in humans. While bacterial contamination in meat has been widely studied, fungal contamination remains overlooked due to limited evidence of immediate disease and the perception that its risks are lower than those of bacteria, which may contribute to insufficient research, awareness, and standardised surveillance protocols. This review compiles published data on the occurrence of fungi in slaughterhouses over the past twenty-five years. It highlights the primary mould and yeast isolated species, mainly identified based on morphological and microscopic characteristics, providing context for their role in meat safety and occupational health. The findings emphasise the need for improved risk assessment and fungal monitoring in meat plants. Standardised fungal detection and control protocols are also suggested for implementation to enhance meat safety and workplace conditions.

## 1. Introduction

Fungi are ubiquitous microorganisms that can endure as filamentous multicellular forms with hyphae or as unicellular yeasts [1,2,3]. Moulds are filamentous fungi with hyphae connected to several spore-forming structures, such as conidia [2]. In contrast to moulds, yeasts are single-celled microorganisms that do not produce secondary toxic metabolites [4]. Although some species may divide by fission, yeasts reproduce primarily by budding [4].

The high-humidity environment of slaughterhouses is prone to mould growth. Fungal growth and spore production are influenced by water activity and the availability of nutrients such as proteins, carbohydrates, and lipids [2,5].

Several mould species are known to produce mycotoxins and secondary toxic metabolites with teratogenic, mutagenic, and carcinogenic potential due to their ability to interfere with RNA synthesis and to cause DNA damage [2,6,7]. *Aspergillus* spp., *Fusarium* spp., and *Penicillium* spp. are prominent producers of these adverse mycotoxins [6].

The most commonly detected mycotoxins in food products include aflatoxins, chiefly produced by *Aspergillus flavus*, *Aspergillus parasiticus*, and *Aspergillus nomius*; ochratoxin A, mainly produced by *Aspergillus ochraceus*, *Aspergillus niger*, *Aspergillus carbonarius*, and *Penicillium verrucosum*; zearalenone, primarily associated with *Fusarium graminearum*; fumonisins, predominantly originating from *Fusarium verticillioides*, *Fusarium proliferatum*, and *Aspergillus niger*; and deoxynivalenol, generally produced by *Fusarium graminearum* and *Fusarium culmorum* [8]. Mycotoxins are commonly associated with cereals, as they are the most frequently contaminated products [5]. However, they can also be detected in animal-derived products such as meat, eggs, and milk [5].

Among the most toxic mycotoxins in meat, ochratoxin A (OTA) is the primary contaminant, while aflatoxin B1 (AFB1) is less frequent and found in lower concentrations [9]. Other mycotoxins may also be present, but their impact on meat safety remains unclear [10]. More specifically, a water activity range ≥0.80 and temperatures of 2–35 °C may allow the production of mycotoxins, such as AFB1 and OTA, in meat products [9,10].

Meat can be contaminated with mould during the animal production phase and at the slaughterhouse due to improper handling, processing, and equipment contamination. This poses a potential public health risk for consumers since the ingestion of the mycotoxins, produced by certain moulds, can lead to diseases in humans, ranging from fatal outcomes to chronic disruptions in the nervous, cardiovascular, pulmonary, endocrine, and digestive systems [11]. Moreover, yeasts, such as *Candida* spp., may behave as opportunistic pathogens and are associated with foodborne illness in immunocompromised patients [12]. Therefore, monitoring the fungal load during slaughter is essential to ensure meat safety [13].

From an occupational health perspective, chronic exposure or inhalation of mould and its metabolites is reportedly linked to asthma, dermatitis, allergies, respiratory infections, and other infectious diseases in humans [14]. Although fungal infections can affect healthy individuals, immunocompromised individuals are at a higher risk [3].

Besides moulds, yeasts, although less studied, have also been isolated from slaughterhouse lines and equipment and may contribute to opportunistic infections, particularly in immunocompromised individuals [15,16]. However, their role in such infections within the slaughterhouse environment remains inadequately researched and poorly understood.

Preventing fungal dissemination in slaughterhouses is crucial to minimising the adverse effects and mitigating risks for slaughterhouse workers and meat consumers [17].

Nevertheless, microbial monitoring in slaughterhouses typically prioritises bacterial contamination and, due to the lack of research and specific monitoring protocols and guidelines for fungi, fungal contamination still goes unnoticed. This could be attributed to the perception that, although fungi play a significant role in food spoilage, they are less relevant as foodborne pathogens compared to bacteria [18]. This underestimation may lead to unaddressed health risks. Considering this, the present review aims to provide an overview of the occurrence of moulds and yeasts in slaughterhouses based on the published data from the past twenty-five years and discuss their potential impact on meat safety and occupational health, emphasising the urgent need for more research and more fungal monitoring in slaughterhouses.

## 2. Impact of Fungi on Meat Safety and Occupational Health

Globally, more than 300 million people suffer from severe fungal disease, which can result in over 3.8 million deaths per year [19,20].

The routes of fungal infections through exposure to spores may include inhalation, ingestion of contaminated food, and skin contact [3]. As moulds can produce mycotoxins, fungal-related health risks expand beyond direct infections.

The production of mycotoxins is influenced by intrinsic factors (e.g., species and strain) and external conditions, such as humidity, temperature, pH, gas composition, and the nature of the growth substrate [21].

Depending on the dose and exposure duration, the toxic effects of mycotoxins can be acute or chronic, often influencing protein, fat, and carbohydrate metabolism and, consequently, nucleic acid synthesis, which may lead to kidney and liver damage or even cancer [22].

Although the primary route of mycotoxin exposure is the oral ingestion of contaminated food [23], which may pose a significant food safety risk, exposure can also occur through inhalation and dermal contact, representing an occupational hazard and which effect may be more harmful than oral exposure [23].

### 2.1. Fungi as Meat-Borne Pathogens

Meat-borne pathogens consist of more than just bacteria, viruses, and parasites. Fungi can also be present in meat and meat product contaminants, releasing mycotoxins into these products, which can have potentially serious implications for meat safety and public health [24,25].

Fungal spoilage of meat products is typically characterised by the presence of black, white, or blue-green colonies on the surface [24].

The occurrence of moulds in meat, influenced by factors such as temperature (10–45 °C), pH (1.5–10), and water activity (≥0.6) [10], is considered an indicator of the level of hygiene during processing activities [26,27]. *Cladosporium* spp. have been linked to black spot spoilage in dry-cured meats; *Chrysosporium pannorum* is associated with the formation of white spots on frozen meat, and *Penicillium expansum* may cause blue-green spots [24]. Yeasts usually cause gas formation and an unpleasant odour [24].

Commonly isolated fungal genera in red meat include *Cladosporium*, *Geotrichum*, *Mucor*, *Rhizopus*, *Sporotrichum*, *Thamnidium*, *Candida*, and *Torulopsis*. In contrast, in poultry meat, *Candida*, *Debaryomyces*, *Rhodotorula*, and *Yarrowia* are more frequently described [24]. Future studies examining the interactions between different fungal species isolated from meat could provide valuable insights into their roles in spoilage and pathogenicity, thereby contributing to a deeper understanding of their impact on meat quality and safety.

The primary sources of carcass contamination include air, water, walls, floors, workers, working surfaces, and equipment [24,28,29]. The abattoir’s design and layout can also influence air currents, contributing to airborne contamination of carcasses and contact surfaces [29]. Moreover, contamination of meat and final meat products by mycotoxins can occur when animals are fed with contaminated feed [9]. Unfortunately, there is an evident lack of studies that focus on documenting the relative importance of animal feed contamination as a source of mycotoxin contamination of meat compared to other sources of contamination within the slaughterhouse.

### 2.2. Fungi as Occupational Hazards

Slaughterhouse workers are exposed to several zoonoses, such as leptospirosis, brucellosis, Q fever, tuberculosis, avian influenza, and Crimean Congo haemorrhagic fever [30]. Occupational exposure to fungal burden has also been assessed and confirmed [17].

The biological risk in slaughterhouses is from direct and indirect contact with animal matter and exposure to bioaerosols [17]. Bioaerosols involve airborne bacteria, viruses, fungi, and their by-products, including mycotoxins [17]. Factors such as season, building materials, age of the facility, and ventilation conditions influence fungal concentrations and diversity. Studies on fungal bioaerosols have identified *Aspergillus*, *Penicillium*, *Stachybotrys*, *Cladosporium*, *Alternaria*, *Trichoderma*, and various yeasts as the most common indoor and outdoor fungi, with potential implications for severe health issues [2,31].

Indoor air quality is crucial for health and well-being, as people inhale approximately 10 m³ of air daily, which can contain bioaerosols originating from individuals, organic dust, stored products, and air circulation through natural or artificial ventilation systems [2]. Moreover, slaughtering and processing facilities ventilation systems serve as additional reservoirs for aerosolising and spreading airborne microorganisms [17]. Among different types of slaughterhouses, poultry facilities have previously exhibited the highest fungal load compared to cattle and mixed swine-cattle slaughterhouses, which can be attributed to greater indoor sources of fungal contamination [17].

Although European regulations mandate the assessment of biological risks in occupational settings, mycotoxins are not widely recognised as a risk factor. This may lead to an underestimation of exposure, as mycotoxins can persist in the environment even after the removal of fungi [32]. Occupational exposure to aflatoxin B1 (AFB1), a potent hepatocarcinogen, has already been reported among poultry slaughterhouse workers [19].

Occupational exposure to mycotoxins is mainly linked to working in poorly ventilated environments and improperly using protective equipment and clothing [8]. As already mentioned, the primary exposure routes to airborne mycotoxins include inhalation and dermal contact, as mycotoxins can be present in airborne particles and dust that either carry these toxins or have been directly contaminated by fungal excretions [23].

Inhalation of mycotoxins can lead to various adverse health effects, including mucous membrane irritation, epithelial damage, endocrine disruption, systemic symptoms such as fever, nausea, fatigue, and immunosuppression and immunotoxicity. Furthermore, exposure has been associated with nephrotoxicity, acute and chronic liver damage, central nervous system impairment, reproductive effects, and carcinogenic potential [21].

Yeasts are also opportunistic hazards. The most frequently occurring yeast species associated with human disease include *Candida albicans*, *Candida tropicalis*, *Candida glabrata*, *Candida parapsilosis*, and *Cryptococcus neoformans* [33].

## 3. Occurrence of Moulds in the Slaughterhouse

The formation and spread of mould is unavoidable in the slaughterhouse environment due to the considerable amounts of water used, the prone conditions to condensation, and the presence of rests of food adhering to surfaces [5,34].

Most indoor fungi grow at 10–35 °C [35]. In general, moulds do not grow below a relative humidity of 80% or below 75% within a temperature range of 5–40 °C [35]. Species such as *Alternaria alternata*, *Aspergillus fumigatus*, *Mucor plumbeus*, *Rhizopus* spp., and *Rhodotorula* spp. grow at activity water levels and equilibrium relative humidity of more than 0.90 and 90%, respectively [35]. On the other hand, *Aspergillus flavus*, *Aspergillus versicolora*, *Cladosporium cladosporioides*, and *Cladosporium herbarum* require water activity levels of 0.80–0.90 and equilibrium relative humidity of 80–90% [35]. Moreover, there are species that cannot grow at water activity levels of less than 0.80 and equilibrium relative humidity of less than 80%, such as *Aspergillus niger*, *Penicillium aurantiogriseum*, *Penicillium brevicompactum*, *Penicillium chrysogenum*, *Penicillium expansum*, and *Penicillium griseofulvum* [35].

However, there is limited research focused on the specific environmental conditions within slaughterhouses. One study on energy assessment in the Portuguese meat industry [36] reported that relative humidity levels in cold rooms and rapid cooling tunnels in slaughterhouses could range from nearly 80% to 90%. Despite these insights, a gap in research on the environmental conditions that favour mould growth in slaughterhouses remains.

Although mould colour is not a reliable criterion for species identification, *Stachybotrys chartarum*, a common indoor mould, can often be recognised by its characteristic black pigmentation. *Cladosporium* typically forms olive green to brown or black colouration, while *Penicillium* is often recognised by its green colonies. *Aspergillus*, another frequent indoor mould, can develop shades of red or gold [37].

The spread of moulds in the slaughterhouse is common, and identifying specific species plays a crucial role in understanding the potential risks to public health and food safety.

### 3.1. Aspergillus spp.

*Aspergillus* spp. are commonly found in soil, decaying vegetation, seeds, and grains [19,38].

Diverse *Aspergillus* species are found in various indoor environments, with some species being considered opportunistic pathogens [19]. *Aspergillus* spp., the leading cause of human mould infections, cause various serious health issues in both immunocompetent and immunocompromised patients.

Both humans and animals can be affected by aspergillosis, a fungal infection caused by *Aspergillus* species. The symptoms may range from respiratory illnesses and allergic reactions to invasive disease, the most severe form, leading to serious complications affecting the lungs, brain, and kidneys [39].

*Aspergillus fumigatus* is the most clinically significant, followed by *A. flavus*, *A. terreus*, and *A. niger* [40,41]. Due to their small size, conidia from the Aspergillus genus can be easily inhaled by exposed individuals, allowing the colonisation of the upper and lower respiratory tract [19].

Aflatoxins, highly toxic, teratogenic, mutagenic, and carcinogenic compounds, are secondary metabolites of the *Aspergillus* species, mainly *Aspergillus flavus* and *Aspergillus parasiticus*, and are classified as Group 1 carcinogens by the International Agency for Research on Cancer (IARC) [7,21]. The main categories include aflatoxins B_1_ and B_2_, G_1_ and G_2_, and M_1_ [7].

In slaughterhouses, non-specific strains of *Aspergillus* spp. were previously detected in 10% and 2% of air sample isolates collected in the hanging (“moving rails”) and evisceration (“gall bladder separation”) areas, respectively, of an Austrian poultry slaughterhouse [42]. Similarly, *Aspergillus* spp. were detected in 18% (*n* = 98 isolates) of isolates in air samples from both cattle and poultry slaughterhouses in a Pakistani study [43]. In a Korean study across five swine slaughterhouses, *Aspergillus* spp. accounted for 6% of the detected fungal genera [5]. Additionally, a study conducted in a Saudi Arabian slaughterhouse identified *Aspergillus* spp. in air samples and equipment, including a saw and a blade, as well as in a laboratory environment [44]. These species were also isolated in an Indian slaughterhouse [45].

#### 3.1.1. *Aspergillus flavus*

*Aspergillus flavus* produces the aflatoxins B_1_ and B_2_ [7]. Previous studies have documented occupational exposure to AFB_1_, a potent hepatocarcinogen, among poultry slaughterhouse and poultry farm workers [19].

Viegas et al. [32] assessed the aflatoxin B1(AFB_1_) exposure levels of slaughterhouse workers using a serum biomarker that measured both free AFB_1_ and AFB_1_ bound to albumin, providing insight into both recent exposure (acute) and potential chronic exposure (1–2 months earlier). The significantly higher concentrations of AFB_1_ in workers compared to the control group (*p* < 0.0001) support the accuracy of this method in reflecting exposure levels within the slaughterhouse [32]. The authors [32] also acknowledge that urinary biomarkers, such as AFM1 and Aflatoxin-N7-guanine, are more reliable for confirming exposure, as they provide more precise quantitative relationships and reflect biologically effective doses, linking directly to health outcomes, including hepatocellular carcinoma. These biomarkers are considered less invasive than blood sampling and are therefore suggested for future occupational exposure studies [32].

The contamination of chicken meat with the *Aspergillus* species, especially *Aspergillus flavus*, is one of the most hazardous microbial contaminants [27]. In Austria, *Aspergillus flavus* was detected in less than 1% of air samples collected in the hanging area of a poultry slaughterhouse [42].

The isolation of *Aspergillus flavus* from different locations within the abattoir and beef may lead to mycotoxin production in beef meat [29]. In a Serbian study in two beef slaughterhouses, the fungus was recovered from 17% of isolates found in air samples and 18% in floor samples [28]. More recently, in a Nigerian ruminant slaughterhouse, *Aspergillus flavus* accounted for 17.6% of the airborne fungal isolates (*n* = 24 isolates recovered from the skin/hoof burning site, *n* = 18 isolates from the slaughter ground, *n* = 13 isolates from the lairage site, and *n* = 8 isolates) from the meat stall [29].

Similarly, in an Iraqi slaughterhouse, *Aspergillus flavus* was isolated from 17.4% (*n* = 4 isolates) of indoor air sample isolates and 12.3% (*n* = 7 isolates) of isolates in outdoor air samples [46]. At a Nigerian slaughterhouse, the same species was found in 40% of the fungal isolates on slab surfaces (*n* = 10 isolates) [47]. Additionally, it is highlighted that studies focusing on the role of flies (*Musca domestica*) as vectors revealed that fifteen isolates of *Aspergillus flavus* were isolated from flies *(Musca domestica*) collected from slaughterhouses in Saudi Arabia [48], fifteen isolates in Iraq [49], and fifteen isolates from an Irani slaughterhouse [50].

Although the available data provide an overview of the presence of *Aspergillus flavus* in slaughterhouses, methodological differences may introduce bias in the estimation of actual environmental contamination, so caution is required when interpreting the reported percentages. Future studies with standardised protocols and complementary identification approaches could provide a more robust and comparable estimate of *Aspergillus flavus* prevalence in these environments.

#### 3.1.2. *Aspergillus fumigatus*

*Aspergillus fumigatus* produces several immunosuppressive mycotoxins, including gliotoxin, fumagillin, helvolic acid (fumigacin), fumitremorgin A, and Asp-hemolysin, and may be the causative agent of respiratory symptoms such as asthma, allergic sinusitis, cough, and bronchial hyperresponsiveness [19].

*Aspergillus fumigatus* is the primary etiologic agent of invasive aspergillosis, a disease with high mortality rates among immunocompromised individuals [19,41].

Regarding poultry slaughterhouses, *Aspergillus fumigatus* was previously isolated from 3.1% (*n* = 50 CFU/m^3^) of isolates recovered from air samples collected in a Portuguese study [17]; from 8% and 2% of isolates detected in air samples collected from the evisceration and hanging areas, respectively, in an Austrian slaughterhouse [42]; and in samples collected from air and inhalable dust in the turkey evisceration site of an Italian slaughterhouse [51]. Moreover, nine isolates of *Aspergillus fumigatus* were also recovered from flies (*Musca domestica*) collected in an Iraqi slaughterhouse [49].

#### 3.1.3. *Aspergillus ochraceus*

*Aspergillus ochraceus* was previously isolated from 5.5% (*n* = 8 CFU/m^3^) of air samples isolates collected in a large animal slaughterhouse in Portugal [17]. Six isolates of this species were also identified in flies from an Iraqi slaughterhouse [49]. *Aspergillus ochraceus* is recognised as an essential food pathogen, widely distributed and responsible for the production of ochratoxin A, classified as a Group 2B carcinogen by the IARC, meaning it is “possibly carcinogenic to humans” [8,21,52]. Additionally, this species can produce penicillic acid, dihydropenicillic acid, and viomellein [52]. Its toxicological significance is further accentuated by its association with Balkan endemic nephropathy, a disorder linked to consuming food contaminated with penicillic acid and ochratoxin A [52].

#### 3.1.4. *Aspergillus parasiticus*

*Aspergillus parasiticus* is responsible for the production of aflatoxin B1, aflatoxin B2, aflatoxin G1, and aflatoxin G2, classified as Group 1 carcinogens by the IARC, meaning they are carcinogenic to humans [8,21,53].

Four isolates *of Aspergillus parasiticus* were identified in samples obtained from flies (*Musca domestica*) in an Iraqi slaughterhouse [49].

#### 3.1.5. *Aspergillus niger*

*Aspergillus niger* is ubiquitous and known to be the causative pathogen of the so-called “black mould disease” [54]. Although it is reported to produce ochratoxin A and fumonisins (e.g., fumonisin B2)—classified as Group 2B carcinogens (possibly carcinogenic to humans)—*Aspergillus niger* is, in comparison with other filamentous fungi, relatively harmless [8,21,55]. Still, *Aspergillus niger* is considered an opportunistic pathogen and can induce lung and ear infections, particularly in immunocompromised individuals [55].

*Aspergillus niger* was previously isolated from air samples collected in air handling units, as well as in the turkey cutting and evisceration areas of an Italian poultry slaughterhouse [51]. Additionally, it was identified in 35.1% (*n* = 20 isolates) of air sample isolates from the surrounding outdoor environment of an Iraqi slaughterhouse [46] and in 24% (*n* = 6 isolates) of isolates in slab surface samples from a Nigerian slaughterhouse [47]. In another Nigerian study, in a slaughterhouse for cattle and small ruminants, *Aspergillus niger* accounted for 15.1% of the total collected fungal species (*n* = 15 isolates from the slaughter ground, *n* = 12 isolates from the meat stall, *n* = 9 isolates from the lairage site samples, and *n* = 18 isolates from the skin/hoof burning site’s samples) [29]. In flies (*Musca domestica*) collected from slaughterhouses, twenty-one isolates of *Aspergillus niger* were detected in an Iraqi study [49], ten isolates in a Saudi Arabian slaughterhouse [48], and ten isolates in Iran [50].

#### 3.1.6. *Aspergillus terreus*

*Aspergillus terreus* is an increasingly perceived opportunistic fungus with a worldwide distribution, capable of producing many secondary metabolites and mycotoxins [56]. Moreover, it is regarded as one of the most prevalent airborne spores among the *Aspergillus* species, and it is reported to cause allergenic bronchopulmonary aspergillosis [57].

*Aspergillus terreus* accounted for 30% of the isolates (*n* = 30,000 CFU/m^2^) recovered from floor samples in a large Portuguese animal slaughterhouse [17]. Similarly, a study on two Serbian slaughterhouses detected them in 32% of wall samples, 8% of floor samples, and 2% of air samples [28].

#### 3.1.7. Other Isolated *Aspergillus* spp.

Other species of *Aspergillus* have also been isolated at slaughterhouses. *Aspergillus penicilloides* was isolated in an Italian poultry slaughterhouse from air samples collected in air handling units, turkey cutting and evisceration sites, and from inhalable dust collected from a turkey evisceration site [51]. *Aspergillus clavatus* was detected in 2% of isolates in floor samples collected from Serbian beef slaughterhouses [28]. *Aspergillus carneus* and *Aspergillus candidus* were already isolated from a poultry slaughterhouse in Italy [51].

The frequency and locations of isolation of these *Aspergillus* species provide initial insights but are insufficient to assess their impact on meat safety and occupational health. Further studies are needed on their viability in meat, mycotoxin production, worker exposure, and the link to occupational diseases or meat contamination.

#### 3.1.8. Eurotium (Reclassified as *Aspergillus*) spp.

*Eurotium* spp., now renamed *Aspergillus* [58] and generally recognised as benign fungi [58], were also isolated in poultry slaughterhouses [42,51].

*Eurotium* spp., *Eurotium erbarum*, and *Eurotium chevalieri* were reported in air samples collected in two Italian poultry slaughterhouses [51]. Moreover, it was also possible to isolate *Eurotium* spp. from air samples collected from the hanging area in an Austrian slaughterhouse, and they represented 9% of the isolates obtained in the mentioned area [42].

### 3.2. Fusarium spp.

*Fusarium* species, highly adaptable fungi, are found in various habitats, such as in the air, water, soil, and in plants and organic substrates [59,60]. Some species act as opportunistic pathogens, originating a wide range of infections with high morbidity and mortality [59]. In addition, they produce diverse mycotoxins, such as fumonisins, zearalenone, and trichothecenes, which can be harmful or even fatal to humans and animals [59,61].

*Fusarium* spp. were previously isolated from air handling units in an Italian poultry slaughterhouse [51], and in the evisceration area (5%) and hanging area (less than 1%) of an Austrian poultry slaughterhouse [42]. Additionally, in each of the two studies analysing isolates from flies (*Musca domestica*) in slaughterhouses, five isolates of *Fusarium* spp. were detected in the samples [48,50].

*Fusarium oxysporum,* a human pathogen [62], was detected in 10.9% of the total isolated airborne fungal species from a Nigerian ruminant slaughterhouse (*n* = 12 isolates from the skin/hoof burning site, *n* = 11 isolates from the slaughter ground, *n* = 8 isolates from the lairage site, and *n* = 8 isolates from the meat stall samples) [29]. *Fusarium oxysporum* can produce fumonisin, zearalenone, the neurotoxic fusaric acid, and several cytotoxic mycotoxins such as beauvericin, diacetoxyscirpenol, enniatins, and moniliformin [62].

Additionally, an Iranian study analysing samples collected from the external body surface of flies (*Musca domestica*) in slaughterhouse and hospital environments identified *Fusarium proliferatum,* one of the main species producing fumonisins [63], in 1% of the isolates (*n* = 5 isolates) from the slaughterhouse [50].

### 3.3. Penicillium spp.

*Penicillium* spp. are among the most common filamentous fungi in the food processing industry [64]. It can also proliferate on building walls, substantially in environments with high humidity [64]. This genus plays a significant role in various fields, including food spoilage, biotechnology, plant biology, and medicine [65].

*Penicillium* spp. were already isolated in poultry, cattle, and swine slaughterhouses [5,17,28,29,42,46,47,49,50,51].

*Penicillium* spp. were widely isolated in two Italian poultry slaughterhouses in several processing areas [51]. In a Portuguese study, *Penicillium* spp. were isolated from 32.8% of the species recovered from air samples collected in a poultry slaughterhouse (*n* = 524 isolates). A Pakistani study concluded that 29.3% of the isolates detected in air samples (*n* = 160 isolates) were *Penicillium* spp. [43]. In Austria, 11% of the fungal species isolated from the hanging area and 10% of those isolated in the evisceration site of a poultry slaughterhouse were classified as belonging to *Penicillium* spp. [42].

In a ruminant slaughterhouse, 14.5% of the isolates belonged to *Penicillium* spp., being more isolated from the lairage site samples (*n* = 21 isolates) than from other sites (*n* = 14 isolates from skin/hoof burning site samples, *n* = 12 isolates found in the slaughter ground, and *n* = 5 isolates detected in the meat stall) [29].

A Portuguese study detected 118 CFU/m^3^ (80.8% of the isolates) and 270 CFU/m^3^ (33.3% of the isolates) of airborne *Penicillium* spp. in a large animal slaughterhouse and a mixed beef–swine slaughterhouse, respectively [17]. *Penicillium* spp. also represented 15% of the detected genera in studied Korean swine slaughterhouses [5].

In Nigeria, seven isolates (28% of the fungal isolates) found on slab surfaces were identified as *Penicillium* spp. [47]. The same genus was isolated from samples collected from blades in a Saudi Arabian slaughterhouse [44]. Al-Fattly [46] isolated six Penicillium spp. from inside air samples (26.1% of the indoor fungi isolates) and nine from outdoor samples (15.8% of the outdoor fungi isolates).

Furthermore, slaughterhouse flies (*Musca domestica*) were found to be vectors of four Penicillium isolates in a study conducted in Saudi Arabia [48].

*Penicillium verrucosum* produces ochratoxin A, a Group 2B carcinogen, which is also nephrotoxic, hepatotoxic, and immunotoxic, with the potential to cause chromosomal aberrations in human lymphocytes [64,66]. Seven isolates of *Penicillium verrucosum* were previously isolated from flies (*Musca domestica*) collected from a slaughterhouse in Iraq [49].

*Penicillium notatum* (reclassified as *Penicillium rubens*) [67] accounted for 26.1% and 26.3% of the isolates in indoor and outdoor samples, respectively (*n* = 6 isolates and *n* = 15 isolates, respectively) [46].

In a study on flies (*Musca domestica*) found in slaughterhouse environments, eighteen isolates of *Penicillium aurantiogriseum* were identified [49].

In a Serbian study conducted in two beef slaughterhouses, *Penicillium brevicompactum* represented 19% of isolates in air samples, 8% in floor samples, and 2% in wall samples [28]. In the same facilities, *Penicillium chrysogenum* (reclassified as *Penicillium rubens*) [67] accounted for 5% of the isolates from air samples, 7% from floor samples, and 3% from walls [28]. *Penicillium solitum* was also isolated, comprising 34% of the air sample isolates and 28% of the floor sample isolates [28].

Moreover, *Penicillium polonicum* and *Penicillium expansum* were also identified in a study conducted in two Italian slaughterhouses [51].

### 3.4. Mucor spp.

*Mucor* is one of the largest genera within the order Mucorales [68,69]. Species of this genus are predominantly saprotrophic and found in several environments [68].

*Mucor* spp., reportedly able to produce mycotoxins [70], can cause mucormycosis, a spectrum of opportunistic human infections, varying from chronic cutaneous to rhinocerebral forms [71]. However, there is an unfortunate gap in studies examining the mycotoxin-producing capabilities of different mould species, particularly *Mucor* spp., in slaughterhouse environments.

In a recent Nigerian study, *Mucor* spp. represented 5.3% of the isolated fungal airborne species in a ruminant slaughterhouse, being isolated from the skin/hoof burning site (*n* = 3 isolates), from lairage site samples (*n* = 7 isolates), from samples from the meat stall (*n* = 2 isolates), and from the slaughter ground (*n* = 7 isolates) [29]. Although the low percentage may suggest a limited presence of *Mucor* spp., it is essential to consider that the different isolation locations in the Nigerian study suggest that these moulds may be widely dispersed throughout the slaughterhouse environment, potentially increasing the risk to both meat safety and occupational health.

Similarly, *Mucor* spp. were isolated from 13.2% (*n* = 72 isolates) of air samples’ fungal isolates collected in a Pakistani study [43]—and, in an Iraqi slaughterhouse, from 30.4% and 10.5% of the isolates collected from indoor and outdoor samples, respectively (*n* = 7 isolates and *n* = 6 isolates, respectively) [46]. Additionally, *Mucor* spp. were the only ones found in samples collected from the floor of a Portuguese poultry slaughterhouse [17].

Specific species of *Mucor* isolated from slaughterhouses included *Mucor racemosus*, which can infect humans [72], found in 9% of the isolates recovered from floor samples and 6% of air samples collected in a study carried out in two beef slaughterhouses [28]. *Mucor plumbeus* was detected in air handling unit samples in a study carried out in Italy, and *Mucor circinelloides*, described as being involved in infections [71], was represented by nine isolates collected from slaughterhouse flies [49].

Among these species, *Mucor racemosus* and *Mucor circinelloides* appear to pose a greater risk to human health due to their known association with infections [73], with *Mucor circinelloides* one of the most frequent species within Mucorales causing fatal mucormycosis [74].

Although *Mucor* is a ubiquitous genus, future research should focus on identifying potential sources of contamination within slaughterhouses, which would help improve the understanding of the associated risks.

### 3.5. Rhizopus spp.

*Rhizopus* spp. are also opportunistic fungi responsible for mucormycosis, a rare but life-threatening infection that essentially affects immunocompromised individuals [75,76,77]. Mucormycosis is an emerging opportunistic fungal disease that involves rhino–orbital–cerebral, pulmonary, cutaneous, gastrointestinal, renal, and disseminated forms [77].

*Rhizopus* spp. were previously isolated in 5.9% of samples from a ruminant slaughterhouse (*n* = 9 isolates in samples from the skin/hoof burning site, *n* = 7 isolates in samples from lairage sites, *n* = 3 isolates in meat stall samples, and *n* = 2 isolates in samples from the slaughter ground) [29].

*Rhizopus stolonifer*, a saprophytic fungus that thrives across various temperature and humidity conditions, with an optimal growth temperature of 25 °C [78], was identified in 5.1% of the isolates (*n* = 8) recovered from flies in an Iraqi study [49].

### 3.6. Cladosporium spp.

Black moulds of *Cladosporium,* a ubiquitous genus, are among the most common fungi in indoor and outdoor environments [79]. Some species have clinical significance and can cause allergies [80].

Regarding presence in poultry slaughterhouses, *Cladosporium* spp. were identified in an Italian study, specifically in inhalable dust and air samples collected from turkey cutting and evisceration sites, as well as in air handling unit samples [51]. Likewise, *Cladosporium* spp. were isolated in 26% and 43% of samples from hanging and eviscerating areas, respectively [42]. In a cattle and small ruminant slaughterhouse, this fungal genus was recovered from the skin/hoof burning site (*n* = 5 isolates), lairage site (*n* = 3 isolates), meat stall (*n* = 10 isolates), and slaughter ground (*n* = 4 isolates), constituting 6.1% of the isolates [29]. Additionally, a Portuguese study concluded that 5.5% of the isolates recovered from a cattle slaughterhouse’s air samples and 45.7% of the isolates in air samples from a mixed swine–cattle slaughterhouse were also *Cladosporium* spp. (*n* = 8 CFU/m^3^ and *n* = 370 CFU/m^3^, respectively) [17]. In Korean swine slaughterhouses, *Cladosporium* spp. represented 20% of the detectable genera [5]. A Pakistani study reported that this genus comprised 14.3% of isolates from air samples (*n* = 78 isolates) [43]. *Cladosporium* spp. were also detected in a Saudi Arabian slaughterhouse [44].

Over the past twenty-five years, a small range of studies has reported the isolation of *Cladosporium herbarum,* namely, from various air and inhalable dust samples [28,51]. In Serbian cattle slaughterhouses, *Cladosporium herbarum* was isolated in 25% of wall samples, 4% of floor samples, and 2% of air samples [28]. Other species of *Cladosporium* were also found in the aforementioned slaughterhouses [28]. *Cladosporium sphaerospermum* and *Cladosporium cladosporioides* were also present in walls, floor, and air samples [28].

### 3.7. Alternaria spp.

Black moulds of the genus *Alternaria* commonly infest damp buildings [81]. They can produce potentially harmful food contaminants, known as *Alternaria* toxins, which have been reported to act as mycotoxins [81,82]. *Alternaria* is also considered a potent sensitising aeroallergen, being strongly associated with respiratory disorders such as asthma, rhinosinusitis, pneumonitis, skin infections, and bronchopulmonary mycosis [83].

*Alternaria* spp. were previously isolated from an Austrian poultry slaughterhouse, representing 4% and 7% of the isolates obtained from hanging area and eviscerating area air samples, respectively [42]. In a Pakistani research work, *Alternaria* spp. accounted for 12.7% of air sample isolates (*n* = 69 isolates) [43]. Meanwhile, in swine slaughterhouses, *Alternaria* spp. were described as representing 1% of the detected genera [5]. In an Indian study, *Alternaria* spp. were among the isolated fungal species [45].

Analysis of the vector role of flies (*Musca domestica*) in slaughterhouses has identified *Alternaria* spp. as one of the fungal species present in the external body samples of these flies. A Saudi Arabian study identified two isolates [48], similar findings to those of an Irani study [50], and a third study found twelve isolates of *Alternaria alternata* in samples from flies collected in an Iraqi slaughterhouse [49].

*Alternaria alternata* was also identified in a Serbian study in floor (7% of the isolates) and air samples (2% of isolates) [28].

### 3.8. Botrytis spp.

*Botrytis* spp. are found worldwide but have a low prevalence in both indoor and outdoor ambient air, though they possess allergic potential [84]. Based on peer-reviewed studies published in the last twenty-five years, *Botrytis* spp. were only isolated in poultry slaughterhouses studied in Italy—where *Botrytis cinerea* was identified in inhalable dust collected in a turkey evisceration site and in air samples from air handling units and turkey cutting and evisceration sites [51]—and in Austria—where *Botrytis* spp. represented less than 1% of isolates detected in air samples from the hanging area [42].

### 3.9. Geotrichum spp.

*Geotrichum* spp. were isolated from poultry (less than 1% in eviscerating area air samples) [42] and ruminant slaughterhouses (4.5% of the isolates) [29]. In a Nigerian ruminant slaughterhouse, nine isolates of *Geotrichum* spp. were obtained from slaughter ground samples, four isolates from the meat stall, and three isolates from the skin/hoof burning site, totalling 4.5% of the isolates in that study [29].

Although *Geotrichum*-related diseases are rare, they may be linked to pulmonary and disseminated infections in immunocompromised individuals [85]. In an Italian study, *Geotrichum candidum*, an opportunistic pathogen linked with high mortality rates among cancer patients [85], was identified in air samples from two poultry slaughterhouses and inhalable dust from one of them [51].

### 3.10. Scopulariopsis spp.

*Scopulariopsis* spp. are moulds linked to clinical manifestations most often associated with pulmonary and disseminated infections [86]. *Scopulariopsis,* particularly *Scopulariopsis brevicaulis*, is recognized as a cause of invasive and non-invasive infections, including onychomycosis, keratitis, conjunctivitis, endocarditis, and disseminated infections in both animals and humans [87]. Coupled with the nonspecific clinical manifestations of *Scopulariopsis* infections, limited reports on occupational infections or meat contamination make it challenging to assess its impact on occupational health and meat safety in slaughterhouses due to the lack of specific transmission data in this context.

*Scopulariopsis* spp. were identified in less than 1% of the isolates recovered from air samples in the hanging area site in a poultry slaughterhouse in Austria [42]. In contrast, *Scopulariopsis* spp. were isolated in 59.5% of air samples collected from a Portuguese poultry slaughterhouse (*n* = 950 CFU/m^3^) [17].

Although neither study discusses the possible association between location and isolated species, the sampling location may have influenced the differences between the studies. Taking into account the data obtained in the Portuguese study [17], the overall airborne fungal load was more significant in the reception and bleeding areas, in contrast to the low isolated general fungal load in the hanging and evisceration areas, the only areas studied in the Austrian study [42]. In fact, the reception and bleeding areas revealed levels above the limits recommended by the World Health Organization (WHO), with a maximum value of 150 CFU/m^3^ [17].

Regarding cattle slaughterhouses, *Scopulariopsis brumpti* was detected in 40% of the isolates (*n* = 40,000 CFU/m^2^) collected from floor samples in a Portuguese study [17] and *Scopulariopsis brevicaulis* was similarly isolated from floor samples in a Serbian study, representing 2% of the floor samples’ isolates [28].

Exposure to *Scopulariopsis* spp. in the workplace, positive examination findings, and suspected asthma risk, were identified as key variables in assessing occupational health risks [88]. The likelihood of diagnosis for individuals working in environments where both *Scopulariopsis* spp. and *Cladosporium* spp. were present was 2.01 times higher than for those not exposed to these fungi [88]. Additionally, *Scopulariopsis brevicaulis* has been linked to occupational allergies and onychomycosis [89]. Therefore, it can be suggested that slaughterhouse workers may face an occupational risk of infection from *Scopulariopsis*. However, currently, no studies have confirmed this, and further research is needed to investigate this potential risk.

### 3.11. Other Moulds Isolated in Slaughterhouses

Other moulds with occupational hazard potential isolated in slaughterhouses include *Pseudogymnoascus* spp., which represented 2% of the isolates in five Korean swine slaughterhouses [5]; *Trichoderma* spp., obtained in less than 1% of air samples from the hanging area of a poultry slaughterhouse [42]; *Ulocladium chatarum*, which was isolated from air samples in air handling units and sites for the evisceration and cutting of turkeys in an Italian slaughterhouse [51]; *Wallemia* spp., detected in 4% of the isolates of the poultry eviscerating area air samples [42]; and *Zygomycetes* spp., isolated from inhalable dust samples collected in another two poultry slaughterhouses [51].

Furthermore, *Acremonium* spp. were isolated in two studies conducted in poultry slaughterhouses [42,51]. *Aureobasidium* spp. were obtained in 11.1% of isolates in air samples from a mixed swine–cattle slaughterhouse in Portugal (*n* = 90 CFU/m^3^) [17]. *Aureobasidium pullulans* was recovered from air samples collected in the turkey evisceration site of an Italian poultry slaughterhouse [51].

*Microsporum* spp. were isolated in a Saudi Arabian slaughterhouse [44]. Flies (*Musca domestica*) in slaughterhouses were previously reported as carriers of *Nannizzia gypsea* (formerly *Mycrosporum gypseum* [48,50].

*Monascus ruber* [51], *Chrysonilia* spp., *Cunninghamella* spp. [42], *Emmericella* spp. [42,51], *Englyodontium album* [51], *Epicoccum* spp. [51], *Geomyces* spp. [5,51], *Phoma* spp. [42,51], and *Talaromyces* spp. [51] are moulds also isolated in slaughterhouses.

## 4. Occurrence of Yeasts in the Slaughterhouse

Yeasts are microorganisms found in various environments, including food production facilities such as slaughterhouses [15]. Their presence in these settings may result in contamination of equipment, air, and surfaces throughout the processing chain. While some yeasts are considered harmless, others can act as opportunistic pathogens.

### 4.1. Candida spp.

*Candida* species are part of the mucous flora and can cause a broad scope of human infections. The incidence of infections caused by *Candida* genus has increased significantly in the last decades [90]. *Candida* species are responsible for most human infections caused by fungal pathogens [91]. Opportunistic *Candida* spp. infections, including foodborne illness [12], pose a significant threat to immunocompromised individuals [92]. *Candida albicans* and *Candida parapsilosis* can cause invasive candidiasis in humans, and some of their strains can be transmitted through contaminated food [93]. Also, their biofilm formation capabilities in food processing facilities may contribute to the recurring contamination of meat products [93].

In a Nigerian study, 5.6% of the isolates in a ruminant slaughterhouse belonged to *Candida* spp. [29]. These were also detected in a Saudi Arabian study [44].

More specifically, *Candida albicans* was isolated from 23.3% and 16.7% of the isolated yeasts in broiler carcasses and workers’ swabs, respectively, collected in two Egyptian poultry slaughterhouses [94]. Moreover, in two Iraqi sheep slaughterhouses, *Candida albicans* was isolated in 10% of sheep organs (*n* = 10 isolates), 8% of equipment swabbed before the slaughter process (*n* = 4 isolates), and 12% of equipment swabbed after slaughter (*n* = 6 isolates) [15]. *Candida albicans* was also one of the fungal species previously described as being carried by flies (*Musca domestica*) present in an Iraqi slaughterhouse (*n* = 11 isolates) [49]. *Candida albicans*, a commensal organism, is part of the microbiota in healthy individuals. However, under certain conditions, it can transition from a commensal to a pathogenic state [95]. Considering the given information, although there has been no research focus on this issue in the slaughterhouse context, immunocompromised workers may be at greater risk; the importance of good hygiene practices and effective cleaning and disinfection protocols in the slaughterhouse is emphasised.

Another recognised opportunistic pathogen is *Candida tropicalis*, which is considered the most prevalent pathogenic yeast species within the *Candida* non-albicans group [96]. In humans, *Candida tropicalis* is associated with superficial mycoses, such as onychomycosis, otomycosis, oral and skin candidiasis, keratitis, and genital tract infections [97]. *Candida tropicalis* was reported in swabs from broiler carcasses (16.7%, *n* = 5 isolates) [94], from sheep organs (1%, *n* = 1 isolate) [15], and from flies (4.5%, *n* = 7 isolates) [49].

*Candida parapsilosis*, one of the most prevalent species responsible for candidemia worldwide [98], was isolated in broiler carcass swabs (13.3%, *n* = 4 isolates) and in swabs from workers’ hands (6.7%, *n* = 2 isolates), in a study carried out in two poultry slaughterhouses located in Egypt [94]. Ref. [15] also isolated *Candida parapsilosis* in swabs from sheep organs (4%, *n* = 4 isolates) and from equipment before (8%, *n* = 4 isolates) and after slaughter (2%, *n* = 1 isolate).

Other isolated *Candida* strains consist of *Candida lusitaniae*, detected in two poultry slaughterhouses and in two sheep slaughterhouses [15,94]; *Candida famata*, isolated in an Italian poultry slaughterhouse and in two Iraqi sheep slaughterhouses [15,51]; *Candida gulliermondi* and *Candida zeylanoides*, detected in sheep organs and equipment swabs [15]; *Candida rugosa*, obtained from equipment swabs in sheep slaughterhouses [15]; *Candida sphaerica*, found in sheep organ samples [15]; and *Candida minuta*, found in air samples from an Italian slaughterhouse [51].

The interpretation of the results from the various studies mentioned should take into account the differences in sample types and environmental conditions. As previously suggested, future studies should implement a standardised approach to facilitate a more reliable comparison of fungal occurrence and frequency in slaughterhouses.

### 4.2. Cryptococcus spp.

Cryptococcosis encompasses infections caused by species within the genus *Cryptococcus*, usually leading to pneumonia and/or meningitis [99]. *Cryptococcus* spp. are ubiquitous, non-motile yeasts found worldwide, affecting both humans and animals [100,101].

*Cryptococcus* spp. were isolated from knife, blade, and saw swabs in a Saudi Arabian slaughterhouse [44]. Additionally, *Cryptococcus albidus* was isolated from broiler carcasses (10%, *n* = 3 isolates) [94] and equipment swabbed before slaughter (12%, *n* = 6 isolates) [15].

*Cryptococcus albidus* (syn. *Naganishia albida*) rarely causes infections in humans, but it can act as an opportunistic pathogen in immunocompromised patients. This saprophytic yeast exhibits pathogenic potential by enhancing its resistance to host defences [82,102].

### 4.3. Rhodoturula spp.

Human fungal infections caused by *Rhodotorula* spp. have been increasing over the last few decades, and these are considered emerging pathogens that primarily affect immunocompromised individuals [103]. Reports indicate that *Rhodotorula* spp. can cause fungaemia, meningitis, cutaneous infections, peritonitis, keratitis, ventriculitis, and other less common conditions [103].

*Rhodotorula* spp. were previously isolated from air samples in two Italian poultry slaughterhouses [51]. *Rhodoturula* spp. were also recovered from broiler carcass swabs (26.7%, *n* = 8 isolates) in an Egyptian study, and represented 6% of the detectable genera isolated in a study carried out in Korean swine slaughterhouses [5].

*Rhodotorula mucilaginosa* (formerly *Rhodotorula rubra*) was detected in swabs from equipment taken before (18%, *n* = 9 isolates) and after slaughter (12%, *n* = 6 isolates) [15]. According to the study [15], no significant differences were observed in the isolation proportions of yeasts before and after the slaughter process. However, the fact that *Rhodotorula* spp. are associated with biofilm formation [104], including *Rhodotorula mucilaginosa* [105], and were detected at both sampling points suggests that biofilms may contribute to the resistance and persistence on equipment. This could indicate that the cleaning and disinfection procedures are not sufficient to eliminate the organism or that these techniques are not being appropriately applied.

This strain was also recovered from air samples collected in the cutting site of a turkey slaughterhouse [51] and from blade swabs [44].

*Rhodotorula mucilaginosa*, although rarely seen as an opportunistic pathogen, is one of the most common causative species of fungemia and may be associated with endocarditis in patients with chronic kidney disease [103,106]. Therefore, awareness of opportunistic fungal infections is relevant for occupational health surveillance, especially among vulnerable workers.

Other *Rhodotorula* isolates include *Rhodotorula aurantiaca* and *Rhodotorula minuta*, both of which were isolated from air samples collected at an Italian poultry slaughterhouse [51]. *Rhodotorula minuta*, along with *Rhodotorula mucilaginosa,* is recognised as an opportunistic pathogen [107], whereas the pathogenicity of *Rhodotorula aurantiaca* remains less understood.

Although there are no studies specifically focusing on the environmental conditions in slaughterhouses that favour the growth of *Rhodotorula* spp., this is a ubiquitous species capable of withstanding extreme environments [108]. Therefore, it is possible that this species may have adapted to the varying temperatures found in slaughterhouses.

### 4.4. Other Yeasts Isolated in Slaughterhouses

Various yeast genera have been isolated in slaughterhouses, with studies from different countries highlighting their presence in poultry and swine slaughterhouses. In a Korean study, *Apiotrichum* spp. (3% of the detected genera), *Naganishia* (2% of the detectable genera), *Cystobasidium* spp. (7% of the detected genera), and *Filobasidium* spp. (2% of the detected genera) were isolated [5].

In an Egyptian study, *Torulopsis* spp. were identified in 6.7% of the isolated yeast species in broiler carcass swabs (*n* = 2 isolates) [94]. *Saccharomyces* spp. were identified under the same circumstances [94].

## 5. Microbiological Monitoring in Slaughterhouses: The Overlooked Impact of Fungi

*Penicillium* spp. [28,43,46], *Aspergilus flavus* and *Aspergillus niger* [29], *Cladosporium* spp. [5,42], *Scopulariopsis* spp. [17], *Candida* spp. [94], and *Rhodotorula mucilaginosa* [15] were the most frequently isolated fungal species from slaughterhouses in the limited research carried out over the last twenty-five years. This could present a risk to both food safety and occupational health. However, there is a notable absence of risk assessments and regulations addressing the potential dangers associated with fungi and their secondary metabolites.

Most of the studies addressing the isolation of moulds and yeasts in the slaughterhouse are outdated and, in addition to being scarce, often describe the isolation of fungal species in slaughterhouses without establishing a statistical correlation with other factors, such as isolation sites, slaughterhouse type (species slaughtered, size, etc.), indoor humidity, ventilation, disinfection practices, and more. There is also a lack of studies conducted in slaughterhouses regarding the presence of mycotoxins in meat and the surrounding processing environment.

Toxic residues in animal products mainly arise from contaminated feed. Mycotoxins can be excreted in urine, faeces, and milk, or accumulate in eggs, meat, and internal organs, with aflatoxins predominating in the liver, gizzard, kidney, milk, eggs, and meat [109]. These products are significant sources of human mycotoxin intake [109]. As described in this review, meat contamination may also occur during slaughter. However, the current microbiological monitoring protocols in slaughterhouses do not include fungi or mycotoxins.

According to European legislation, point 3.2 in chapter 3 of the Commission Regulation (CE) No. 2073/2005 [110] states that *Salmonella*, *Enterobacteriaceae*, bacterial aerobic colony count, and *Escherichia coli* must be sampled and monitored in the context of slaughterhouses. It thus does not mention fungal species. Commission Regulation (CE) No. 2023/915 [111], which sets maximum levels for specific contaminants in food, does not set a maximum level for mycotoxins in meat.

Monitoring mycotoxigenic fungi could be an added component of HACCP programs in the meat production chain, and mould removal should be incorporated into the Standard Sanitary Operating Procedures (SSOPs) of slaughterhouses and meat production companies.

Furthermore, based on the data collected in this review, the role of possible vectors in slaughterhouses, such as flies and other insects, should be evaluated.

Although the available studies do not allow for a reliable comparison, slaughterhouse workers may be exposed to lower risk in comparison to those in other food industries and industrial settings, such as grain processing and feed production, who tend to frequently and directly handle highly contaminated raw materials [112]. As mentioned, Viegas et al. [32] confirmed aflatoxin B1 occupational exposure levels in a poultry slaughterhouse, although they were lower than those found in poultry production. The authors [32] explain that this may be because the chickens are slaughtered at the beginning of the process and, after that, workers in most workplaces wear gloves when handling the birds, and thus not all environments in the slaughterhouses present conditions conducive to exposure. Furthermore, strict hygienic conditions applied in this environment can help eliminate and prevent the contamination and spread of fungi, something impossible to guarantee in poultry production.

Regarding occupational exposure, among the fungal species mentioned in this review, only *Aspergillus* spp. (including *Aspergillus flavus* and *Aspergillus fumigatus*), *Candida parapsilosis*, and *Candida tropicalis* are included in the list relating to the protection of workers against risks related to exposure to biological agents at work, published in the Commission Directive (EU) 2000/54 [113], emended by Commission Directive (EU) 2019/1833 [114]. These species are classified as Group 2 biological agents, meaning that the agents can cause diseases in humans and pose a risk to workers. Still, the probability of spreading in the community is low and, generally, there are effective means of prophylaxis or treatment [113].

Moreover, although the most recent report published by the European Agency for Safety and Health at Work [30] mentions the possibility of occupational exposure to fungi, such as histoplasmosis and cryptococcosis, it does not provide detailed insight into the exposure of slaughterhouse workers to other fungal species not associated with direct or indirect contact with animals and their fluids.

Currently, there is no specific regulation in the European Union that addresses mycotoxins as an occupational risk. Previous studies, including those by Viegas et al. [17] and Viegas et al. [32], have highlighted the exposure of slaughterhouse workers to various fungal species and the mycotoxin aflatoxin B1, revealing a significant gap in the comprehensive assessment of the occupational burden caused by fungal exposure in slaughterhouses across Europe. Despite this, there remains a lack of studies focused on assessing the risk of mycotoxin exposure among slaughterhouse workers.

Mycotoxin exposure requires the presence of mould and favourable conditions for toxin production, with humidity being a key factor [115]. However, exposure can occur even without visible mould, as small fungal biomass may produce significant mycotoxins [115]. However, there is a notable gap in the study of environmental factors in slaughterhouses that may contribute to mycotoxin production by moulds and how these conditions may influence meat safety and occupational health.

Assessing occupational risks involves considering the concentration of airborne mycotoxins, exposure duration, and frequency. Two main scenarios exist: regular low-level exposure or occasional high peaks. Health risk evaluation remains challenging due to the absence of regulatory limits for airborne mycotoxins [115]. The lack of monitoring of these compounds in workplaces makes it challenging to compare exposure levels, highlighting the need for standardized methods.

According to Viegas et al. [112], the key aspects that should be considered when occupational exposure assessment to mycotoxins is planned or performed include the recognition of the possible presence of mycotoxins in the workplace, the identification of possible exposure/contamination sources, the collection of contextual information, the characterization of exposure variability, and the definition of Similar Exposure Groups (SEGs).

Airborne fungi are often used as indirect indicators of mycotoxins [116]. However, in addition to the unreliability of this approach, as mycotoxins can persist even after fungi are eliminated [116], there is also a lack of standardisation in fungal identification in slaughterhouses. Alternative methods for assessing mycotoxin exposure also present limitations. PCR may over- or underestimate contamination, and ELISA-based biomonitoring cannot distinguish occupational from dietary exposure [21]. Thus, a more accurate assessment of occupational risks related to airborne mycotoxin exposure requires an integrated approach that combines different sampling and laboratory methods (culture-based, PCR, High-Performance Liquid Chromatography, ELISA, cytotoxicity tests, etc.) [21].

The lack of epidemiological studies makes it difficult to assess the acute and chronic health effects of mycotoxin exposure and to establish regulatory occupational exposure limits for individual mycotoxins and their mixtures [116]. Given the World Health Organisation (WHO) guidelines on airborne fungal spore concentrations, correlating spore counts with mycotoxin presence could help establish risk thresholds. Considering these guidelines, airborne fungal spore concentrations above 500 CFU/m^3^ are considered hazardous, and those above 1000 CFU/m^3^ may be extremely hazardous [117].

Protective measures should be taken to reduce potential exposure to fungal burden. Preventive measures should be implemented, such as disinfection and the use of personal protection devices, including filtration masks and gloves [17].

Increased awareness and education on this topic should be provided to both workers and employers. During worker training sessions, topics covered may include good hygiene practices, cleaning and disinfection protocols and, among other potential occupational health risks, those associated with occupational exposure to mycotoxins. Also, explaining and increasing awareness regarding possible meat contamination sources would be beneficial for meat safety in the slaughterhouse context. Mothershaw et al. [118] previously suggested that demonstrations conducted in small groups, in the workers’ native language, ensure better understanding among slaughterhouse workers undergoing training. Avoiding written instructions and using interactive methods and pictorial posters strategically placed can enhance participation and aid in reinforcing key messages [118].

Ultimately, it is important to highlight that the presence of fungi capable of producing mycotoxins in slaughterhouse wastewater poses an environmental risk [119].

## 6. Conclusions

Fungal contamination in slaughterhouses remains an overlooked issue. This fact may pose implications for meat safety and occupational health. However, current microbiological monitoring protocols and regulatory frameworks do not address the presence of mycotoxigenic fungi in such an environment. Specific guidelines for assessing and mitigating these risks are still lacking, maybe due to the gap in research and available data. Further research would be essential to establish the limits of mycotoxins in meat and to assess occupational exposure limits, as well as to evaluate the long-term health impacts of mycotoxin exposure in the slaughterhouse.

Furthermore, implementing regular and planned standardised fungal monitoring programs in air and surface samples, integrating mould control and cleaning into Standard Sanitary Operating Procedures (SSOPs), and raising awareness among workers, employers, and regulatory entities about the issue are essential steps to enhance public health. Establishing mycotoxin limits in meat and addressing occupational exposure should be a priority. This should be based on toxicological studies, sensitive analytical techniques, and epidemiological assessments to define regulatory standards, thereby ensuring food safety and protecting worker health.

## Data Availability

No new data were created or analysed in this study. Data sharing is not applicable to this article.

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
