# Peer review of "Occurrence of Moulds and Yeasts in the Slaughterhouse: The Underestimated Role of Fungi in Meat Safety and Occupational Health"

_foods, 2025, doi:10.3390/foods14081320_

Round 1

Reviewer 1 Report

Comments and Suggestions for Authors

Dear Authors,

Overall, this review article is well-written, well-organized and easy to understand. The chosen references are appropriate.

It would be great to insert a section, relating to the optimal conditions for the growth and reproduce of molds and yeasts. Report in details any conditions that contribute to molds and yeasts growth.

Line 563-564: references are not well mentioned.

Author Response

EDITOR COMMENTS AND SUGGESTIONS TO THE AUTHOR

Answer to referee’s comments and suggestions.

The authors are grateful to the reviewers and the editor for their attentive and detailed comments, which significantly contributed to the improvement of the review paper.

We hope that the answers below and the modifications introduced in the manuscript are clear and concise enough to meet the requirements of the Reviewer, enabling the publication of the manuscript.

We have incorporated the modifications as follows. All the revisions are marked in the text. Changes derived from reviewers' recommendations are underlined in yellow, while changes in English are marked in green.

DETAILED RESPONSE TO REVIEWER 1

GENERAL COMMENTS TO THE AUTHOR

Reviewer´s comment:

Overall, this review article is well-written, well-organized and easy to understand. The chosen references are appropriate.

It would be great to insert a section, relating to the optimal conditions for the growth and reproduce of molds and yeasts. Report in details any conditions that contribute to molds and yeasts growth.

Our reply: Excellent suggestion. We completely agree. Although there is data on environmental conditions in the building that favour mould growth, there is no concrete data on relative humidity, temperature, or water activity inside the slaughterhouse. However, we will add a paragraph on the ideal environmental conditions for mould growth and mycotoxin production, mentioning the conditions previously studied in meat products. In addition, we believe it is essential to highlight the importance of examining the environmental conditions of slaughterhouses, which can contribute to understanding the ecological conditions of both fungi and other pathogens.

We added the following statement:

Most indoor fungi grow at 10–35 °C [x - WHO, 2009]. In general, moulds do not grow below a relative humidity of 80% or below 75% within a temperature range of 5–40 °C [x - WHO, 2009]. Species such as Alternaria alternata, Aspergillus fumigatus, Mucor plumbeus, Rhizopus spp., and Rhodotorula spp. grow at activity water levels and equilibrium relative humidity of more than 0.90 and 90%, respectively [x - WHO, 2009]. On the other hand, Aspergillus flavus, Aspergillus versicolora, Cladosporium cladosporioides, and Cladosporium herbarum require water activity levels between 0.80–0.90 and equilibrium relative humidity of 80–90% [x - WHO, 2009]. Moreover, they are species that can not grow at water activity levels of less than 0.80 and equilibrium relative humidity of less than 80%, such as Aspergillus niger, Penicillium aurantiogriseum, Penicillium brevicompactum, Penicillium chrysogenum, Penicillium expansum, and Penicillium griseofulvum [x - WHO, 2009].

However, there is limited research focused on the specific environmental conditions within slaughterhouses. One study on energy assessment in the Portuguese meat industry [x – Nunes et al., 2016] reported that relative humidity levels in cold rooms and rapid cooling tunnels in slaughterhouses could range from nearly 80% to 90%. Despite these insights, a gap in research on the environmental conditions that favour mould growth in slaughterhouses remains. 

Reviewer´s comment:

Line 563-564: references are not well mentioned.

Our reply: Indeed. Thank you very much for this point. The references are not well mentioned. We will proceed with the correction.

Reviewer 2 Report

Comments and Suggestions for Authors

Comments to author:

This article offers a novel perspective on fungal prevention in slaughterhouses, presenting a thorough analysis of the collected data and concluding with valuable insights. The manuscript is well-organized and provides a comprehensive examination of the data, making significant contributions to the fields of public health and food safety. To further enhance the manuscript and its impact, the authors may consider addressing the following issues:

Abstract

line 18 – 19: “While bacterial meat contamination has been extensively studied, fungal contamination remains overlooked due to insufficient research, awareness, and standardised surveillance protocols.”

Why is fungal contamination overlooked compared to bacterial meat contamination, and is this explanation sufficient?

line 21 – 22: “It aims to highlight the primary mould and yeast isolated species, providing a context on their role in meat safety and occupational health.”

Are the primary mould and yeast isolated species clearly identified?

  1. Introduction

lines 43 – 50: Is the information about the occurrence of mycotoxins in cereals and plant products sufficient to draw a valid comparison with their presence in meat and meat products?

lines 52 – 54: Is the link between fungal contamination in slaughterhouses and public health risks for consumers adequately substantiated?

lines 59 – 61: Is the role of yeasts in causing opportunistic infections in slaughterhouses clearly defined and backed by relevant research?

lines 63 – 65: Are the reasons for the lack of focus on fungal monitoring in slaughterhouses, such as the lack of research and specific protocols, thoroughly explored?

  1. Impact of Fungi on Meat Safety and Occupational Health

Line 108: What is the relative importance of animal feed contamination as a source of mould in meat compared to other sources mentioned?

lines 89 – 108: How do environmental factors within the slaughterhouse (e.g., temperature, humidity) affect the growth and prevalence of different fungal genera on meat?

lines 89 – 108: Are there any known interactions between different fungal species isolated from meat that could impact their spoilage potential or pathogenicity?

  1. Occurrence of moulds in the slaughterhouse

lines 148 – 149: Are the conditions described (use of water, condensation, food rests) the only significant factors contributing to mould formation and spread in the slaughterhouse environment?

lines 150 – 153: Are the visual characteristics used to identify the mentioned mould species (colour of colonies) reliable and sufficient for accurate identification in a slaughterhouse setting?

lines 188 – 189: Are the levels of aflatoxin B₁ and B₂ produced by Aspergillus flavus accurately quantified and reported in the relevant studies?

lines 196 – 208: Are the percentages of Aspergillus flavus isolation in different studies truly representative of the overall contamination levels, considering potential sampling biases?

lines 270 – 276: Are the isolation locations and frequencies of these other Aspergillus species sufficient to assess their potential impact on meat safety and occupational health?

lines 270 – 276: What are the specific characteristics or potential risks associated with these other Aspergillus species isolated in the slaughterhouse environment?

lines 353 – 357: Are the mycotoxin-producing capabilities of Mucor spp. well-documented and quantified in the context of slaughterhouse environments?

lines 358 – 361: How do the different isolation locations (e.g., skin/hoof burning site, lairage site, etc.) in the Nigerian study relate to the potential risks of Mucor spp. to meat safety and occupational health?

lines 368 – 372: Are there any differences in the pathogenicity or virulence of the specific Mucor species isolated (e.g., Mucor racemosus, Mucor plumbeus, Mucor circinelloides) in the slaughterhouse context?

lines 358 – 373: What are the potential sources of Mucor spp. contamination in the slaughterhouse environment?

lines 448 – 452: Are the clinical manifestations of Scopulariopsis spp. infections well-documented and specific enough to assess their potential impact on meat safety and occupational health in the slaughterhouse context?

lines 454 – 455: What could be the reasons for the significant difference in the prevalence of Scopulariopsis spp. between the Austrian and Portuguese poultry slaughterhouses?

lines 455 – 458: How do the high prevalence rates of Scopulariopsis spp. in the Portuguese poultry slaughterhouse air samples and cattle slaughterhouse floor samples relate to the potential risks for meat contamination and worker health?

lines 448 – 460: What factors might contribute to the relatively low prevalence of Scopulariopsis spp. in the Austrian poultry slaughterhouse compared to other studies?

lines 448 – 460: Are there any known control measures or strategies to prevent or reduce the presence of Scopulariopsis spp. in slaughterhouses?

  1. Occurrence of yeasts in the slaughterhouse

lines 486 – 508: Are the methods used to detect Candida spp. in these various studies (Nigerian, Saudi Arabian, Egyptian, Iraqi) reliable and comparable, considering the different sample types and environments?

lines 501 – 502: How does the ability of Candida albicans to transition from a commensal to a pathogenic state impact the risk assessment for meat safety and occupational health in the slaughterhouse context?

lines 486 – 508: How do the findings regarding Candida spp. in this study fit into the broader understanding of the role of fungi in meat safety and occupational health in slaughterhouses? (lines 486 - 508)

lines 533 – 551: What are the specific environmental factors in the slaughterhouses that might be contributing to the presence and isolation of Rhodotorula spp.?

lines 543 – 545: How does the presence of Rhodotorula mucilaginosa on equipment before and after slaughter affect the risk of contamination to the meat?

lines 547 – 549: Given that Rhodotorula mucilaginosa is associated with endocarditis in chronic kidney disease patients, what are the implications for workers' health in the slaughterhouse?

lines 550 – 551: How do the different Rhodotorula species isolated (e.g., Rhodotorula aurantiaca, Rhodotorula minuta) compare in terms of their pathogenicity and potential risks in the slaughterhouse setting?

  1. Microbiological monitoring in slaughterhouses: the overlooked impact of fungi

lines 617 – 618: What are the specific humidity levels and other environmental conditions within the slaughterhouse that are most conducive to mycotoxin production by moulds?

lines 620 – 622: How can we more accurately assess occupational risks associated with mycotoxin exposure, especially considering the lack of regulatory limits for airborne mycotoxins?

lines 623 – 624: What would be the most effective standardized methods for monitoring mycotoxins in the workplace, given the current challenges?

lines 625 – 627: How reliable are the alternative methods to using airborne fungi as indirect indicators of mycotoxins, considering the persistence of mycotoxins after fungi elimination?

lines 634 – 635: How can we effectively increase awareness and education on mycotoxin exposure risks among workers and employers in the slaughterhouse setting?

lines 617 – 637: How do the risks associated with mycotoxin exposure in the slaughterhouse compare to those in other food production or industrial settings?

  1. Conclusion

lines 646 – 648: How can standardized fungal monitoring programs be effectively designed and implemented in the unique context of slaughterhouses? (lines 646 - 648)

What are the most appropriate methods for establishing the limits of mycotoxins in meat and occupational exposure limits?

Comments on the Quality of English Language

 The English could be improved

Author Response

EDITOR COMMENTS AND SUGGESTIONS TO THE AUTHOR

Answer to referee’s comments and suggestions.

The authors are grateful to the reviewers and the editor for their attentive and detailed remarks, which helped considerably improve the review paper.

We hope the answers below and modifications introduced in the manuscript are clear and concise enough as required by the Reviewer to enable the publication of the manuscript.

We have incorporated the modifications as follows. All the revisions are marked in the text. Changes derived from reviewers' recommendations are underlined in yellow, while changes in English are marked in green.

DETAILED RESPONSE TO REVIEWER 2

GENERAL COMMENTS TO THE AUTHOR

  1. Reviewer´s comment:

This article offers a novel perspective on fungal prevention in slaughterhouses, presenting a thorough analysis of the collected data and concluding with valuable insights. The manuscript is well-organized and provides a comprehensive examination of the data, making significant contributions to the fields of public health and food safety. To further enhance the manuscript and its impact, the authors may consider addressing the following issues.

Our reply:  Thank you for your thoughtful and encouraging feedback. We sincerely appreciate your recognition of our work and its contributions to public health and food safety. Your suggestions are highly valuable, and we hope that we have addressed the mentioned issues to further improve the manuscript.

Abstract

  1. Reviewer´s comment:

Line 18 – 19: “While bacterial meat contamination has been extensively studied, fungal contamination remains overlooked due to insufficient research, awareness, and standardised surveillance protocols.”

Why is fungal contamination overlooked compared to bacterial meat contamination, and is this explanation sufficient?

Our reply: Fungal contamination is often overlooked due to a lack of research and standardized monitoring protocols. The research focus has predominantly been on bacteria, given their more immediate impact on food safety, while fungi, although they can also be harmful, have not been as well studied or recognized due to this gap. In recent years, few studies have focused on fungal contamination in meat in the context of slaughterhouses and these few do not provide relevant information such as, for example, association with the slaughter of animals fed with mycotoxin-contaminated food, association of the presence of different fungal species with specific environmental factors of the slaughterhouse (condensation level, relative humidity, temperature, water activity, etc.), the mycotoxin production capabilities of the different isolated species, differences in pathogenicity or virulence between different species in the context of slaughterhouses, association with possible sources of contamination and, much less, discuss options for monitoring programs for moulds and yeasts in the context of slaughterhouses.

The reasons behind this lack of scientific investment may relate to lower immediate health risks compared to bacteria and limited evidence of direct human infections from meat-borne fungi. Also, fungal surveillance is not mandated in standard meat hygiene checks, except for visible spoilage moulds in extreme cases and the absence of regulatory limits for fungal contamination in meat products reduces industry incentives to monitor it.

Therefore, the main objective of this review is to alert the scientific community and health authorities to this issue, in an attempt to combine efforts to increase monitoring and risk assessment research, in order to truly understand the real impact of fungi on the health of end consumers and slaughterhouse workers, in addition to the fact that this type of research would contribute to setting a maximum exposure limit.

We corrected the line pointed out to:

While bacterial contamination in meat has been widely studied, fungal contamination remains overlooked due to limited evidence of immediate disease and the perception that its risks are lower than those of bacteria, which may contribute to insufficient re-search, awareness, and standardised surveillance protocols.

  1. Reviewer´s comment:

Line 21 – 22: “It aims to highlight the primary mould and yeast isolated species, providing a context on their role in meat safety and occupational health.”

 Are the primary mould and yeast isolated species clearly identified?

Our reply: Most of the studies employed robust methods, including molecular techniques and standard mycological procedures, to clearly identify the isolated fungal species, although Al-Fattly (2013), did not clearly explain species identification due to lack of clarity and homogenized writing. Most have relied primarily on culture-based methods to isolate the fungi. They remain the first choice and may be required by law for the detection of microorganisms in food testing laboratories. However, some limitations include: the inability to culture a wide range of microbial groups (low sensitivity), disadvantages for slow-growing cultures, slow turnaround times, labor-intensive processes, and their unsuitability for the rapid detection of microorganisms (Aladhadh, 2023).

According to Aladhadh (2023), culture-based methods, when combined with advanced techniques such as MALDI-TOF MS, PCR, and NGS, enhance the rapid detection and identification of foodborne bacterial and fungal pathogens. While fungi and their mycotoxins can be ingested through contaminated food, culture-dependent methods require up to seven days for fungal growth. MALDI-TOF MS has been successfully used to identify key fungal groups like Rhizopus, Aspergillus, Fusarium, and Mucor in various foods, though its application to fungi remains limited due to the lack of curated spectral databases. ELISA and PCR-ELISA have been developed for detecting pathogenic fungi and mycotoxins, such as aflatoxins in stored foods. Despite its effectiveness, PCR-based methods are less frequently reported for fungal detection in food compared to bacteria. Similarly, while NGS is less commonly applied to fungal pathogens, it has been used for their detection in clinical and environmental samples.

We added the following statement:

It highlights the primary mould and yeast isolated species, mainly identified based on morphological and microscopic characteristics, providing context for their role in meat safety and occupational health.

Introduction

  1. Reviewer´s comment:

Lines 43 – 50: Although mycotoxins are more extensively described in cereals and other plant products, they are also found in meat and meat products [5]. In food, the most fre-quently detected mycotoxins consist of aflatoxins, chiefly produced by Aspergillus fla-vus, Aspergillus parasiticus, and Aspergillus nomius; ochratoxin A, mainly produced by Aspergillus ochraceus, Aspergillus niger, Aspergillus carbonarius, and Penicillium verru-cosum; zearalenone, primarily associated with Fusarium graminearum; fumonisins, pre-dominantly originated by Fusarium verticillioides, Fusarium proliferatum, and Aspergillus niger; and deoxynivalenol, generally produced by Fusarium graminearum and Fusarium culmorum [8].

Is the information about the occurrence of mycotoxins in cereals and plant products sufficient to draw a valid comparison with their presence in meat and meat products?

Our reply: Although mycotoxins are more extensively described in cereals and other plant products, they are also found in meat and meat products" is intended to highlight the disparity in research attention between cereals/plant products and meat, rather than to suggest a direct comparison between their mycotoxin contamination levels. Furthermore, the paragraph primarily serves to introduce the major foodborne mycotoxins and their fungal sources in a broad food safety context, before narrowing the focus to meat and meat products. The intent is not to equate contamination levels or risk factors between plant-based and animal-based foods, but rather to acknowledge the presence of mycotoxins in both categories while underscoring the relative research gap concerning meat.

The paragraph was altered to:

The most commonly detected mycotoxins in food products include aflatoxins, chiefly produced by Aspergillus flavus, Aspergillus parasiticus, and Aspergillus nomius; ochratoxin A, mainly produced by Aspergillus ochraceus, Aspergillus niger, Aspergillus carbonarius, and Penicillium verrucosum; zearalenone, primarily associated with Fusarium graminearum; fumonisins, predominantly originated by Fusarium verticil-lioides, Fusarium proliferatum, and Aspergillus niger; and deoxynivalenol, generally produced by Fusarium graminearum and Fusarium culmorum [x – Franco et al., 2020]. Mycotoxins are commonly associated with cereals, as they are the most commonly contaminated products [3 – Lee et al., 2024]. However, they can also be detected in animal-derived products such as meat, eggs, and milk [3 – Lee et al., 2024].  

The following paragraph was also added in order to provide more information about the specific occurrence of mycotoxins on meat and meat products, in order to state the difference from common mycotoxins in foods (in general):

Among the most toxic mycotoxins in meat, ochratoxin A (OTA) is the primary contaminant, while aflatoxin B1 (AFB1) is less frequent and found in lower concentra-tions [x - Pleadin et al., 2021]. Other mycotoxins may also be present, but their impact on meat safety remains unclear [x - Zadravec et al., 2019]. More specifically, a range of water activity range ≥0.80 and temperatures between 2–35°C may allow the produc-tion of mycotoxins, such as AFB1 and OTA, in meat products [x - Zadravec et al., 2019; x - Pleadin et al., 2021].

Added references:

Pleadin, J., Lešić, T., Milićević, D., Markov, K., Šarkanj, B., Vahčić, N., ... & Zadravec, M. (2021). Pathways of mycotoxin occurrence in meat products: A review. Processes, 9(12), 2122.

Zadravec, M., Markov, K., Frece, J., Perković, I., Jakopović, Ž., Lešić, T., ... & Pleadin, J. (2019). Toxicogenic fungi and the occurrence of mycotoxins in traditional meat products. Croatian journal of food science and technology, 11(2), 272-282.

  1. Reviewer´s comment:

Lines 52 – 54: Meat can be contaminated with mould during the animal production phase and at the slaughterhouse due to improper handling, processing, and equipment contamina-tion. This poses a potential public health risk for consumers. Therefore, monitoring the fungal load during slaughter is essential to ensure meat safety.

Is the link between fungal contamination in slaughterhouses and public health risks for consumers adequately substantiated? 

Our reply: We agree that the risk of meat contamination in the slaughterhouse context by moulds and yeasts should be better explained. This risk is primarily due to the ability of certain mould species to produce mycotoxins, which are toxic secondary metabolites that can persist in food and pose a serious health risk to consumers. Chronic exposure to specific mycotoxins, such as aflatoxins and ochratoxin A, has been associated with carcinogenic, hepatotoxic, and immunosuppressive effects.

Additionally, some yeasts are known foodborne contaminants that may cause gastric infections, particularly in immunocompromised individuals. Given these risks, we have refined our discussion to clarify why fungal contamination in slaughterhouses is relevant to food safety and public health, emphasizing both the direct impact on consumers and the increased risk for vulnerable populations.

We correct the pointed out lines to:

This poses a potential public health risk for consumers since the ingestion of the myco-toxins, produced by certain moulds, can lead to diseases in humans, ranging from fatal outcomes to chronic disruptions in the nervous, cardiovascular, pulmonary, endocrine, and digestive systems [x – Mahdy, et al., 2019]. Moreover, yeasts, such as Candida spp., may behave as opportunistic pathogens and be associated with foodborne illness in immunocompromised patients [x -Sharma et al., 2017].

Added references:

Mahdy, A., Salem, A., & Zaghloul, M. (2019). Some organic acids as antifungal on frozen duck meat. Benha Veterinary Medical Journal, 37(1), 73-76.

Sharma, K., Chattopadhyay, U., & Naskar, K. (2017). Prevalence of Candida albicans in raw chicken and mutton meat samples sold in the open markets of Kolkata city of West Bengal. Int. J. Livest. Res, 7, 243-249.

  1. Reviewer´s comment:

Lines 59 – 61: Besides moulds, yeasts are also isolated in slaughterhouse lines and equipment and may contribute to opportunistic infections [11, 12].

 Is the role of yeasts in causing opportunistic infections in slaughterhouses clearly defined and backed by relevant research?

Our reply: We acknowledge the importance of better defining the role of yeasts in causing opportunistic infections in slaughterhouses. Although moulds are often the focus of research regarding contamination, yeasts, such as Candida spp., Cryptococcus, and Rhodotorula, have been increasingly recognized as potential causes of opportunistic infections, particularly in immunocompromised individuals.

Recent studies have shown that yeasts can colonize slaughterhouse environments, including equipment, surfaces, and animal carcasses, contributing to cross-contamination.

Despite these findings, the current body of research remains insufficient, and there is a need for more focused studies to fully understand the role of yeasts in foodborne illnesses and occupational health risks in slaughterhouses.

We correct the pointed out lines to:

Besides moulds, yeasts, although less studied, have also been isolated from slaughterhouse lines and equipment and may contribute to opportunistic infections, particularly in immunocompromised individuals [8, Mashari & Al-haddad, 2024; 9, Nakamura et al., 2022]. However, their role in such infections within the slaughter-house environment remains inadequately researched and poorly understood.

  1. Reviewer´s comment:

Lines 63 – 65: Nevertheless, microbial monitoring in slaughterhouses typically prioritises bacterial contamination, and due to the lack of research and specific monitoring protocols and guidelines for fungi, fungal contamination still goes unnoticed.

Are the reasons for the lack of focus on fungal monitoring in slaughterhouses, such as the lack of research and specific protocols, thoroughly explored?

Our reply: We acknowledge the concern regarding the lack of focus on fungal monitoring in slaughterhouses. Although no research focuses on the reasons for the lack of focus on fungal monitoring in slaughterhouses, we may present some reasons. Historically, the primary focus of microbial monitoring has been on bacterial pathogens, such as Salmonella and Escherichia coli, due to their direct association with foodborne illnesses and well-established regulatory standards. This has led to the development of extensive research and well-defined protocols for bacterial contamination in food safety.

In contrast, fungal contamination has received less attention, partly due to a lack of awareness about its potential health risks and the longer incubation period often associated with fungal diseases. Additionally, mycotoxin contamination is a more recent focus in the context of meat safety, and there are still a lack of regulatory frameworks or research studies dedicated to fungal contaminants in slaughterhouses.

As a result, specific protocols and guidelines for fungal monitoring have not been developed to the same extent as those for bacteria, leaving fungal contamination largely unnoticed and under-monitored in slaughterhouse environments. This gap in research and regulation underscores the importance of further studies to address the potential health risks associated with fungal contamination in meat production.

We added the following information to the outlined statement:

This could be attributed to the perception that, although fungi play a significant role in food spoilage, they are less relevant as foodborne pathogens compared to bacteria [x – Anderson et al., 2000].

Added references:

Anderson, J. G., Rowan, N. J., MacGregor, S. J., Fouracre, R. A., & Farish, O. (2000). Inactivation of food-borne enteropathogenic bacteria and spoilage fungi using pulsed-light. IEEE Transactions on Plasma Science, 28(1), 83-88.

Impact of Fungi on Meat Safety and Occupational Health

  1. Reviewer´s comment:

Line 108: However, mould in meat can also result from animals being fed contaminated feed.

What is the relative importance of animal feed contamination as a source of mould in meat compared to other sources mentioned?

Our reply: In fact, we would like to correct moulds to mycotoxins. Although the reference actually uses the term moulds, consulting the reference used by these authors, we confirm that what our reference actually meant to mention is the presence of mycotoxins in meat due to the consumption of feed contaminated by slaughtered animals.

There is no scientific research that determines the relative importance of each possible source of contamination and the sources of contamination in studies that isolated moulds and yeasts in the context of slaughterhouses are not extensively discussed, so it is not possible to objectively discuss the impact of animal feed contamination on mould contamination in meat.

The following statement was added:

Moreover, contamination of meat and final meat products by mycotoxins can occur when animals are fed contaminated feed [x – Pleadin et al., 2021]. Unfortunately, there is a clear lack of studies that focus on documenting the relative importance of animal feed contamination as a source of mycotoxin contamination of meat compared to other sources of contamination within the slaughterhouse.

Added references:

Pleadin, J., Lešić, T., Milićević, D., Markov, K., Šarkanj, B., Vahčić, N., ... & Zadravec, M. (2021). Pathways of mycotoxin occurrence in meat products: A review. Processes, 9(12), 2122.

  1. Reviewer´s comment:

Lines 89 – 108: “Meat-borne pathogens consist of more than just bacteria, viruses, and parasites. Fungi can also be present in meat and meat product contaminants, releasing mycotox-ins into the contaminated products, causing potentially serious implications on meat safety and public health [19, 20]. Fungal spoilage of meat products is typically characterised by the presence of black, white, or blue-green colonies on the surface [19]. The occurrence of moulds in meat, a significant source of food spoilage, is consid-ered an indicator of the level of hygiene during processing activities [21, 22]. Cladosporium spp. have been linked to black spot spoilage in dry-cured meats; Chryso-sporium pannorum is associated with the formation of white spots on frozen meat, and Penicillium expansum may originate blue-green spots [19]. Yeasts usually cause gas formation and an unpleasant odour [19]. Commonly isolated fungal genera in red meat include Cladosporium, Geotrichum, Mucor, Rhizopus, Sporotrichum, Thamnidium, Candida, and Torulopsis. In contrast, in poultry meat, Candida, Debaryomyces, Rhodotorula, and Yarrowia are more frequently described [19]. The primary sources of carcass contamination include air, water, walls, floors, workers, working surfaces, and equipment [19, 23, 24]. The abattoir’s design and lay-out can also influence air currents, contributing to airborne contamination of carcasses and contact surfaces [24]. However, mould in meat can also result from animals being fed contaminated feed [25]”.

How do environmental factors within the slaughterhouse (e.g., temperature, humidity) affect the growth and prevalence of different fungal genera on meat?

Our reply: Although there is data on environmental conditions in the building that favour mould growth, there is no concrete data on relative humidity, temperature, or water activity inside the slaughterhouse. However, we will add a paragraph on the ideal environmental conditions for mould growth and mycotoxin production, mentioning the conditions previously studied in meat products. In addition, we believe it is important to draw attention to the need to study the environmental conditions of slaughterhouses, which can contribute to the study of ecological conditions of both fungi and other pathogens.

The following section was added:

The occurrence of moulds in meat, influenced by factors such as temperature (10–45°C), pH (1.5–10), and water activity (≥0.6) [x - Zadravec et al., 2019], is considered an indicator of the level of hygiene during processing activities [14 – Hussein et al., 2023; 15 – Shaltout et al., 2021].

Moreover, the following statement was added in the section Occurrence of moulds in the slaughterhouse:

Most indoor fungi grow at 10–35 °C [x - WHO, 2009]. In general, moulds do not grow below a relative humidity of 80% or below 75% within a temperature range of 5–40 °C [x - WHO, 2009]. Species such as Alternaria alternata, Aspergillus fumigatus, Mucor plumbeus, Rhizopus spp., and Rhodotorula spp. grow at activity water levels and equi-librium relative humidity of more than 0.90 and 90%, respectively [x - WHO, 2009]. On the other hand, Aspergillus flavus, Aspergillus versicolora, Cladosporium cladosporioides, and Cladosporium herbarum require water activity levels between 0.80–0.90 and equi-librium relative humidity of 80–90% [x - WHO, 2009]. Moreover, they are species that can not grow at water activity levels of less than 0.80 and equilibrium relative humid-ity of less than 80%, such as Aspergillus niger, Penicillium aurantiogriseum, Penicillium brevicompactum, Penicillium chrysogenum, Penicillium expansum, and Penicillium griseo-fulvum [x - WHO, 2009].

However, there is limited research focused on the specific environmental condi-tions within slaughterhouses. One study on energy assessment in the Portuguese meat industry [x – Nunes et al., 2016] reported that relative humidity levels in cold rooms and rapid cooling tunnels in slaughterhouses could range from nearly 80% to 90%. Despite these insights, a gap in research on the environmental conditions that favour mould growth in slaughterhouses remains.    

Added references:

Zadravec, M., Markov, K., Frece, J., Perković, I., Jakopović, Ž., Lešić, T., ... & Pleadin, J. (2019). Toxicogenic fungi and the occurrence of mycotoxins in traditional meat products. Croatian journal of food science and technology, 11(2), 272-282.

World Health Organization. (2009). WHO guidelines for indoor air quality: dampness and mould. In WHO guidelines for indoor air quality: dampness and mould.

Nunes, J., Da Silva, P. D., Andrade, L. P., Domingues, L., & Gaspar, P. D. (2016). Energy assessment of the Portuguese meat industry. Energy Efficiency, 9, 1163-1178.

  1. Reviewer´s comment:

Lines 89 – 108: Are there any known interactions between different fungal species isolated from meat that could impact their spoilage potential or pathogenicity? 

Our reply: To the best of our knowledge and the research carried out to write this review, interactions between fungal species isolated from meat are not widely known and researched. There may possibly be several types of interactions, but they have not been studied much to date.  Therefore, we would like to add a small highlight the need to focus on this issue in future research.

We added the following paragraph:

Future studies examining the interactions between different fungal species isolated from meat could provide valuable insights into their roles in spoilage and pathogenic-ity, thereby contributing to a deeper understanding of their impact on meat quality and safety.

Occurrence of moulds in the slaughterhouse

  1. Reviewer´s comment:

Lines 148 – 149: The formation and spread of mould is unavoidable in the slaughterhouse environment due to the considerable amounts of water used, the prone conditions to con-densation, and the presence of rests of food adhering to surfaces.

Are the conditions described (use of water, condensation, food rests) the only significant factors contributing to mould formation and spread in the slaughterhouse environment?

Our reply: As mentioned, relative humidity, temperature and water activity are relevant environmental factors. Although there is data on environmental conditions in the building that favour mould growth, there is no concrete data on relative humidity, temperature, or water activity inside the slaughterhouse. A statement mentioning these environmental was added, as previously mentioned.

  1. Reviewer´s comment:

Lines 150 – 153: Stachybotrys chartarum, a commonly found indoor mould, is identifiable by its black colour. Cladosporium typically forms olive green to brown or black colouration, while Penicillium is often recognised by its green colonies. Aspergillus, another frequent in-door mould, can develop shades of red or gold [31].

Are the visual characteristics used to identify the mentioned mould species (colour of colonies) reliable and sufficient for accurate identification in a slaughterhouse setting?

Our reply: You are correct that the visual characteristics provided (e.g., the colour of colonies) are not reliable or sufficient for accurate identification of mould species in a slaughterhouse setting. These characteristics serve only as a preliminary guide to potentially identify mould outbreaks and suggest the types of moulds that might be present.

We added the following statement:

Although mould colour is not a reliable criterion for species identification, Stachybotrys chartarum, a common indoor mould, can often be recognised by its char-acteristic black pigmentation. Cladosporium typically forms olive green to brown or black colouration, while Penicillium is often recognised by its green colonies. Aspergil-lus, another frequent indoor mould, can develop shades of red or gold [x – Wahab et al., 2021].

  1. Reviewer´s comment:

Lines 188 – 189: Aspergillus flavus produces the Aflatoxins B₁ and B₂ [7]. Previous studies have documented occupational exposure to AFB₁, a potent hepatocarcinogen, among poultry slaughterhouse and poultry farm workers.

Are the levels of aflatoxin B₁ and B₂ produced by Aspergillus flavus accurately quantified and reported in the relevant studies?

Our reply: Are the occupational levels mentioned by the reviewer? We will assume that these are the levels of aflatoxins that the reviewer is referring to in the question. If this is the correct interpretation of your question, we should mention that only exposure to Aflatoxin B1 has been previously studied in an occupational slaughterhouse context. The opening sentence serves only as an informative note about the type of aflatoxins produced by Aspergillus flavus.

According to Viegas et al., 2016, the levels of aflatoxin B₁ (AFB₁) in the slaughterhouse workers were assessed using a serum biomarker that measured both free AFB₁ and AFB₁ bound to albumin. This method provided insight into both recent exposure (acute) and potential chronic exposure (1–2 months earlier), making it a useful tool for screening both immediate and past exposure to AFB₁. The significant higher concentrations of AFB₁ in workers compared to the control group (P < 0.0001) support the accuracy of this method in reflecting the exposure levels within the slaughterhouse.

While the serum AFB₁ measure is a good exposure indicator, it is not directly related to intake but rather reflects biologically available AFB₁ that has entered the bloodstream. The study also acknowledges that urinary biomarkers such as AFM1 and Aflatoxin-N7-guanine are more reliable for confirming exposure, as they provide clearer quantitative relationships and reflect biologically effective doses, linking directly to health outcomes such as hepatocellular carcinoma. These biomarkers are considered less invasive than blood sampling and are therefore suggested for future occupational exposure studies.

Regarding the exposure route, the study acknowledges that although inhalation might be a significant exposure route, it cannot be solely attributed to airborne particles. The study suggests that in specific areas like the evisceration workplace, dermal absorption could also be contributing to the workers' exposure to AFB₁, as the air contamination was low, and workers in these areas exhibited higher concentrations of AFB₁ despite limited particle contamination.

Thus, the study accurately quantifies the levels of AFB₁ and provides a comprehensive evaluation of exposure through multiple routes, including potential dermal absorption, especially in workers exposed to lower levels of airborne contamination.

We can add a note about the method used by this study and summarize the reliability of the results, as mentioned.

The following statement was added:

Viegas et al. [x Susana Viegas, 2016] assessed the aflatoxin B1(AFB₁) exposure lev-els of slaughterhouse workers using a serum biomarker that measured both free AFB₁ and AFB₁ bound to albumin, providing insight into both recent exposure (acute) and potential chronic exposure (1–2 months earlier). The significantly higher concentra-tions of AFB₁ in workers compared to the control group (P < 0.0001) support the accu-racy of this method in reflecting the exposure levels within the slaughterhouse [x – Susana Viegas et al., 2016]. The authors [x – Susana Viegas et al., 2016] also acknowledge that urinary biomarkers, such as AFM1 and Aflatoxin-N7-guanine, are more reliable for confirming exposure, as they provide clearer quantitative relation-ships and reflect biologically effective doses, linking directly to health outcomes, in-cluding hepatocellular carcinoma. These biomarkers are considered less invasive than blood sampling and are therefore suggested for future occupational exposure studies [x – Susana Viegas et al., 2016].

Added references:

Viegas, S., Veiga, L., Almeida, A., dos Santos, M., Carolino, E., & Viegas, C. (2016). Occupational exposure to aflatoxin B1 in a Portuguese poultry slaughterhouse. Annals of Occupational Hygiene, 60(2), 176-183.

  1. Reviewer´s comment:

Lines 196 – 208: The isolation of Aspergillus flavus from different locations within the abattoir and beef may lead to mycotoxin production in beef meat [24]. In a Serbian study in two beef slaughterhouses, the fungus was recovered from 17% of isolates found in air sam-ples and 18% in floor samples [23]. More recently, in a Nigerian ruminant slaughter-house, Aspergillus flavus accounted for 17.6% of the airborne fungal isolates (n = 24 isolates recovered from the skin/hoof burning site, n = 18 isolates from the slaughter ground, n = 13 isolates from the lairage site, and n = 8 isolates) from the meat stall [24]. Similarly, in an Iraqi slaughterhouse, Aspergillus flavus was isolated from 17.4% (n = 4 isolates) of indoor air samples’ isolates and 12.3% (n = 7 isolates) of isolates in out-door air samples [40]. At a Nigerian slaughterhouse, the same species was found in 40% of the fungal isolates on slab surfaces (n = 10 isolates) [41]. Additionally, it is high-lighted that studies focusing on the role of flies (Musca domestica) as vectors revealed that fifteen isolates of Aspergillus flavus were isolated from flies (Musca domestica) col-lected from slaughterhouses in Saudi Arabia [42], fifteen isolates in Iraq [43], and fif-teen isolates from an Irani slaughterhouse [44].

Are the percentages of Aspergillus flavus isolation in different studies truly representative of the overall contamination levels, considering potential sampling biases?

Our reply: Variations in sampling protocols, culture media used and identification criteria can directly impact the results obtained, making direct comparisons between studies challenging. In addition, the choice of isolation method can affect the sensitivity of detection. Studies that used only culture-based techniques may have underestimated the fungal diversity present, especially if they were not complemented by molecular methods such as PCR or DNA sequencing. The time of exposure of plates to air, the height of collection and the inclusion of environmental samples (such as surfaces and vectors, e.g. flies) also influence the results.

The following statement was added:

Although the available data provide an overview of the presence of Aspergillus flavus in slaughterhouses, methodological differences may introduce bias in the estima-tion of actual environmental contamination, so caution is required when interpreting the reported percentages. Future studies with standardised protocols and complemen-tary identification approaches could provide a more robust and comparable estimate of Aspergillus flavus prevalence in these environments.

  1. Reviewer´s comment:

Lines 270 – 276: Other species of Aspergillus have also been isolated at slaughterhouses. Aspergillus penicilloides was isolated in an Italian poultry slaughterhouse from air samples collected in air handling units, turkeys’ cutting and evisceration sites, and from inhalable dust collected from a turkeys’ evisceration site [45]. Aspergillus clavatus was detected in 2% of isolates in floor samples collected from Serbian beef slaughterhouses [23]. Aspergillus carneus and Aspergillus candidus were already isolated from a poultry slaughterhouse in Italy [45].

 Are the isolation locations and frequencies of these other Aspergillus species sufficient to assess their potential impact on meat safety and occupational health? What are the specific characteristics or potential risks associated with these other Aspergillus species isolated in the slaughterhouse environment?

Our reply: As mentioned above, the representativeness of data on the presence of Aspergillus spp. in slaughterhouses may be influenced by different methodological factors. The frequency of isolation and the locations of detection may provide clues about their presence, but may not be sufficient for a conclusive assessment of their impact on meat safety and occupational health. Additional studies would be needed on the viability of these fungi in meat products, their capacity to produce mycotoxins under real slaughterhouse conditions, and the level of exposure of workers to their spores. Furthermore, the correlation between the presence of these fungi and the occurrence of occupational diseases or meat contamination still needs to be better established.

The following statement was added:

The frequency and locations of isolation of these Aspergillus species provide initial insights but are insufficient to assess their impact on meat safety and occupational health. Further studies are needed on their viability in meat, mycotoxin production, worker exposure, and the link to occupational diseases or meat contamination.

  1. Reviewer´s comment:

Lines 353 – 357: Mucor is one of the largest genera within the order Mucorales [62, 63]. Species of this genus are predominantly saprotrophic and found in several environments [62]. Mucor spp., reportedly able to produce mycotoxins [64], can cause mucormycosis, a spectrum of opportunistic human infections, varying from chronic cutaneous to rhi-nocerebral forms [65].

Are the mycotoxin-producing capabilities of Mucor spp. well documented and quantified in the context of slaughterhouse environments?

Our reply: No, unfortunately there is a clear lack of studies focusing on the capabilities of different species of moulds to produce mycotoxins in the context of slaughterhouses environments, namely Mucor species.

The following statement was added:

However, there is an unfortunate gap in studies examining the mycotoxin-producing capabilities of different mould species, particularly Mucor spp., in slaughterhouse en-vironments.

  1. Reviewer´s comment:

Lines 358 – 361: In a recent Nigerian study, Mucor spp. represented 5.3% of the isolated fungal airborne species in a ruminant slaughterhouse, being isolated from skin/hoof burning site (n = 3 isolates), from lairage site’ samples (n = 7 isolates), from samples of meat stall (n = 2 isolates), and from slaughter ground (n = 7 isolates).

How do the different isolation locations (e.g., skin/hoof burning site, lairage site, etc.) in the Nigerian study relate to the potential risks of Mucor spp. to meat safety and occupational health? 

Our reply: Mucor spp. are ubiquitous fungi capable of colonizing a wide variety of environments, from decaying organic materials, such as dungs, to animal tissues. The different isolation locations of Mucor spp. in the Nigerian study suggest that these moulds may be widely dispersed throughout the slaughterhouse environment, potentially increasing the risk to both meat safety and occupational health.

 The following statement was added:

Although the low percentage may suggest a limited presence of Mucor spp., it is essen-tial to consider that the different isolation locations in the Nigerian study suggest that these moulds may be widely dispersed throughout the slaughterhouse environment, potentially increasing the risk to both meat safety and occupational health.

  1. Reviewer´s comment:

Lines 368 – 372: Specific species of Mucor isolated from slaughterhouses included Mucor racemosus, which can infect humans [66], found in 9% of the isolates recovered from floor samples and 6% of air samples collected in a study carried out in two beef slaughterhouses [23]; Mucor plumbeus, detected in air handling units’ samples in a study carried out in Italy; and Mucor circinelloides, described to be involved in infections [65], represented by nine isolates collected from slaughterhouse’s flies [43].

Are there any differences in the pathogenicity or virulence of the specific Mucor species isolated (e.g., Mucor racemosus, Mucor plumbeus, Mucor circinelloides) in the slaughterhouse context?

Our reply: While all three species have some potential for pathogenicity, Mucor racemosus and Mucor circinelloides appear to pose a greater risk to human health due to their known association with infections.

The following statement was added:

Among these species, Mucor racemosus and Mucor circinelloides appear to pose a greater risk to human health due to their known association with infections [x - Samundi et al., 2022], being the Mucor circinelloides one of the most frequent species within Mucorales causing fatal mucormycosis [x - López-Fernández et al., 2018].

Added references:

López-Fernández, L., Sanchis, M., Navarro-Rodríguez, P., Nicolás, F. E., Silva-Franco, F., Guarro, J., ... & Capilla, J. (2018). Understanding Mucor circinelloides pathogenesis by comparative genomics and phenotypical studies. Virulence, 9(1), 707-720.

Samundi, S. P., Parameswaran, S., Pichaivel, M., & Gopal, M. (2022). An overview of mucormycosis. Innovare J Health Sci, 10(1), 1-7.

  1. Reviewer´s comment:

Lines 358 – 373: What are the potential sources of Mucor spp. contamination in the slaughterhouse environment? 

Our reply: As mentioned, Mucor spp. are ubiquitous fungi capable of colonizing a wide variety of environments. Considering this ecological versatility, potential sources of contamination in the slaughterhouse environment may include the animal tissues themselves during the slaughter process, among others. However, there are no studies specifically investigating the occurrence and origin of Mucor spp. contamination in this context. Therefore, future research should focus on the traceability of Mucor contamination within the slaughterhouse and the identification of its possible sources, contributing to a better understanding of the risks involved.

The following statement was added:

Although Mucor is a ubiquitous genus, future research should focus on identifying potential sources of contamination within slaughterhouses, which would help improve the understanding of the associated risks.

  1. Reviewer´s comment:

Lines 448 – 452: Scopulariopsis spp. are moulds linked to clinical manifestations most often associ-ated with pulmonary and disseminated infections [78]. Scopulariopsis, particularly Scopulariopsis brevicaulis, is recognized as a cause of invasive and non-invasive infections, including onychomycosis, keratitis, conjunctivitis, endocarditis, and disseminated infections in both animals and humans.

Are the clinical manifestations of Scopulariopsis spp. infections well documented and specific enough to assess their potential impact on meat safety and occupational health in the slaughterhouse context?

Our reply: The clinical manifestations of Scopulariopsis infections are nonspecific and can be attributed to different etiological agents. Furthermore, reports of occupational infection or meat contamination by this fungus are limited in the literature. Therefore, the assessment of the impact of Scopulariopsis on occupational health and meat safety in slaughterhouses remains uncertain due to the lack of specific data on its transmission in this context.

We will add a note summarizing the difficulty in assessing the potential impact on meat safety and occupational health in the slaughterhouse context based on currently available research.

The following statement was added:

Coupled with the nonspecific clinical manifestations of Scopulariopsis infections, lim-ited reports on occupational infections or meat contamination make it difficult to as-sess its impact on occupational health and meat safety in slaughterhouses due to the lack of specific transmission data in this context.

  1. Reviewer´s comment:

Lines 454 – 455: Scopulariopsis spp. were identified in less than 1% of the isolates recovered from air samples in the hanging area site in a poultry slaughterhouse in Austria [36]. In contrast, Scopulariopsis spp. were isolated in 59.5% of air samples collected from a Portuguese poultry slaughterhouse (n = 950 CFU/m³).

What could be the reasons for the significant difference in the prevalence of Scopulariopsis spp. between the Austrian and Portuguese poultry slaughterhouses?

Our reply: In an attempt to summarize the main differences between the methods used in the two studies in order to point out a plausible explanation for the differences between the mentioned studies, we present the following data: a) the Portuguese study was carried out in a poultry slaughterhouse with a higher slaughter rate per hour compared to the Austrian study (2500 more chickens/hour); b) the air samples in the Portuguese study were collected in more locations than in the Austrian study. While the Portuguese study collected samples from the hanging area, reception, stacking, bleeding, evisceration and cutting room, the Austrian study collected samples from the hanging area and evisceration; c) the material used in the Portuguese study to collect samples was the Millipore Air Tester, while the Austrian study collected samples from the Anderson Six Stage Viable Cascade Impactor and Bio Sampler; d) the collection and culture methods were similar.

Taking into account the data obtained in the Portuguese study, the airborne fungal load (in general and without specifying isolated species) was more significant in the reception and bleeding areas, in contrast to the low isolated fungal load in the hanging and evisceration areas, the only areas studied in the Austrian study. In fact, the bleeding and reception areas revealed levels above the limits recommended by the World Health Organization (WHO) of maximum value of 150 CFU/m3. Therefore, the sampling location may have influenced and explained the differences between the studies. However, neither study discusses the possible association between location and isolated species.

Therefore, we can only highlight these facts and add the following statement:

Although neither study discusses the possible association between location and isolated species, the sampling location may have influenced the differences between the studies. Taking into account the data obtained in the Portuguese study [x – Viegas et al., 2016], the overall airborne fungal load was more significant in the reception and bleeding areas, in contrast to the low isolated general fungal load in the hanging and evisceration areas, the only areas studied in the Austrian study [x – Haas et al., 2005]. In fact, the bleeding and reception areas revealed levels above the limits recommended by the World Health Organization (WHO) of the maximum value of 150 CFU/m3 [x – Viegas et al., 2016].

  1. Reviewer´s comment:

Lines 455 – 458: Scopulariopsis spp. were identified in less than 1% of the isolates recovered from air samples in the hanging area site in a poultry slaughterhouse in Austria [36]. In contrast, Scopulariopsis spp. were isolated in 59.5% of air samples collected from a Portuguese poultry slaughterhouse (n = 950 CFU/m³) [13]. Regarding cattle slaughter-houses, Scopulariopsis brumpti was detected in 40% of the isolates (n = 40,000 CFU/m²) collected from floor samples in a Portuguese study [13] and Scopulariopsis brevicaulis was similarly isolated from floor samples in a Serbian study, but representing 2% of the floor samples’ isolates [23].

How do the high prevalence rates of Scopulariopsis spp. in the Portuguese poultry slaughterhouse air samples and cattle slaughterhouse floor samples relate to the potential risks for meat contamination and worker health?

Our reply: Several species of Scopulariopsis are also known to cause opportunistic infections and pose an increased health risk to workers. In addition, a previous study on meat products in the context of manufacturing plant, have observed a match between Scopulariopsis isolates in air samples and foodborne isolates. Therefore, we can extrapolate these data to the slaughterhouse context and state that there is a risk of contaminating the meat and that slaughterhouse workers may be exposed to an occupational risk of infections by Scopulariopsis.

The following statement was added:

Exposure to Scopulariopsis spp. in the workplace, along with positive examination findings and suspected asthma risk, were identified as key variables in assessing occu-pational health risks [x – Özdilli et al., 2007]. The likelihood of diagnosing individuals working in environments where both Scopulariopsis spp. and Cladosporium spp. were present was 2.01 times higher than those not exposed to these fungi [x – Özdilli et al., 2007]. Additionally, Scopulariopsis brevicaulis has been linked to occupational allergies and onychomycosis [x – Viegas et al., 2012]. Therefore, it can be suggested that slaugh-terhouse workers may face an occupational risk of infection by Scopulariopsis. Howev-er, currently, no studies are confirming this, and further research is needed to investi-gate this potential risk.

Added references:

Özdilli, K., Işsever, H., Özyildirim, B. A., Hapcioglu, B., Ince, N., Ince, H., ... & Gedikoğlu, G. (2007). Biological hazards in tannery workers. Indoor and Built Environment, 16(4), 349-357.

  1. Reviewer´s comment:

Lines 448 – 460: Scopulariopsis spp. are moulds linked to clinical manifestations most often associated with pulmonary and disseminated infections [78]. Scopulariopsis, particularly Scopulariopsis brevicaulis, is recognised as a cause of invasive and non-invasive infections, including onychomycosis, keratitis, conjunctivitis, endocarditis, and disseminated infections in both animals and humans [79]. Scopulariopsis spp. were identified in less than 1% of the isolates recovered from air samples in the hanging area site in a poultry slaughterhouse in Austria [36]. In contrast, Scopulariopsis spp. were isolated in 59.5% of air samples collected from a Portuguese poultry slaughterhouse (n = 950 CFU/m³) [13]. Regarding cattle slaughter-houses, Scopulariopsis brumpti was detected in 40% of the isolates (n = 40,000 CFU/m²) collected from floor samples in a Portuguese study [13] and Scopulariopsis brevicaulis was similarly isolated from floor samples in a Serbian study, but representing 2% of the floor samples’ isolates [23].

What factors might contribute to the relatively low prevalence of Scopulariopsis spp. in the Austrian poultry slaughterhouse compared to other studies? 

Our reply: It can be stated that both the Austrian and Serbian studies presented contrasting results with the Portuguese study. The Serbian study studied air, water, wall and floor samples and only isolated Scopulariopsis from the floor. However, in comparison with the other two studies (whose materials and methods have already been analyzed previously), this study used a different method for collecting air samples, using the technique of exposing plates of Dichloran-Rosebengal-Chloramphenicol-agar medium to air for 1 minute. All studies identified the fungi based on macroscopic and microscopic characteristics, but using different guidelines, with the Portuguese study using the most up-to-date guideline. As previously mentioned, the Austrian study collected samples from only two areas, areas that correspond to areas with low general fungal load according to the Portuguese study. The Serbian study does not mention the specific locations where samples were collected, so it will be difficult to draw comparisons and point out possible reasons for the differences. Neither study investigates the possible source of contamination and the relationship between sampling location and isolation of Scopulariopsis.

Occurrence of yeasts in the slaughterhouse

  1. Reviewer´s comment:

Lines 486 – 508: Candida species are part of the mucous flora and can cause a broad scope of hu-man infections. The incidence of infections caused by Candida genus has increased sig-nificantly in the last decades [80]. Candida species are responsible for most human infections caused by fungal pathogens [81]. In a Nigerian study, 5.6% of the isolates in a ruminant slaughterhouse belonged to Candida spp. [24]. These were also detected in a Saudi Arabian study [38].  More specifically, Candida albicans was respectively isolated from 23.3% and 16.7% of the isolated yeasts in broiler carcasses and workers’ swabs collected in two Egyptian poultry slaughterhouses [82]. Moreover, in two Iraqi sheep slaughterhouses, Candida albicans was isolated by 10% in sheep organs (n = 10 isolates), by 8% in equipment swabbed before the slaughter process (n = 4 isolates), and by 12% in equipment swabbed after slaughter (n = 6 isolates) [11]. Candida albicans was also one of the fungal species that were previously described to be carried by flies (Musca domestica) present in an Iraqi slaughterhouse (n = 11 isolates) [43]. Candida albicans, a commensal organ-ism, is part of the microbiota in healthy individuals. However, under certain condi-tions, it can transition from a commensal to a pathogenic state [83]. Another recognized opportunistic pathogen is Candida tropicalis, which is consid-ered the most prevalent pathogenic yeast species within the Candida non-albicans group [84]. In humans, Candida tropicalis is associated with superficial mycoses, such as onychomycosis, otomycosis, oral and skin candidiasis, keratitis, and genital tract in-fections [85]. Candida tropicalis was reported in swabs from broiler carcasses (16.7%, n = 5 isolates) [82], from sheep organs (1%, n = 1 isolate) [11], and from flies (4.5%, n = 7 isolates) [43].

Are the methods used to detect Candida spp. in these various studies (Nigerian, Saudi Arabian, Egyptian, Iraqi) reliable and comparable, considering the different sample types and environments?

Our reply: The methods used to detect Candida spp. in these studies appear to be reliable and well-executed, although there are some differences in the sample types and environments. In the studies from Nigeria, Saudi Arabia, Egypt, and Iraq, the use of Sabouraud Dextrose Agar (SDA) supplemented with chloramphenicol to isolate fungi is consistent, which is a standard method for fungal isolation. The identification techniques, including morphological and biochemical tests (such as the germ tube test, urease test, and VITEK 2 system), also demonstrate clarity and accuracy in identifying Candida spp. The studies varied in sample types (e.g., swabs from poultry carcasses, workers' hands, or air samples) and environments (e.g., slaughterhouses and processing plants), which could influence the types and concentrations of fungi isolated. However, the use of additional identification methods such as PCR amplification and RFLP analysis in the Egyptian study adds a higher level of certainty to the identification process. Overall, while the methods differ slightly across the studies, they all appear reliable for detecting Candida spp. in these varied environments. However, we recommend exercising caution when interpreting the results, considering the differences in sample types and environments.

We added the following statement:

The interpretation of the results from the various studies mentioned should take into account the differences in sample types and environmental conditions. As previ-ously suggested, future studies should implement a standardised approach to facilitate a more reliable comparison of fungal occurrence and frequency in slaughterhouses.

  1. Reviewer´s comment:

Lines 501 – 502: Candida albicans, a commensal organism, is part of the microbiota in healthy individuals. However, under certain conditions, it can transition from a commensal to a path-ogenic state [83].

How does the ability of Candida albicans to transition from a commensal to a pathogenic state impact the risk assessment for meat safety and occupational health in the slaughterhouse context? 

Our reply: The pathogenicity of Candida spp. is influenced by the host immune response. Previously, cases of gastroenteritis in immunocompromised patients due to Candida have been reported. Candida infection may also pose a risk to immunocompromised workers. Candida spp. is present as a commensal microorganism in the mucous membranes of healthy animals and humans. Poor hygiene during the slaughter process may result in contamination of meat by workers and exposure of workers to this yeast. In addition, the ability to form biofilms may also contribute to the survival of the yeast on equipment and the environment and lead to cross-contamination.

We added the following statement:

Considering the provided information, although there is no research focus on this issue in the slaughterhouse context, immunocompromised workers may be at greater risk and the importance of good hygiene practices and effective cleaning and disinfection protocols in the slaughterhouse is emphasised.

  1. Reviewer´s comment:

Lines 486 – 508: Candida species are part of the mucous flora and can cause a broad scope of human infections. The incidence of infections caused by Candida genus has increased sig-nificantly in the last decades [80]. Candida species are responsible for most human infections caused by fungal pathogens [81]. In a Nigerian study, 5.6% of the isolates in a ruminant slaughterhouse belonged to Candida spp. [24]. These were also detected in a Saudi Arabian study [38]. More specifically, Candida albicans was respectively isolated from 23.3% and 16.7% of the isolated yeasts in broiler carcasses and workers’ swabs collected in two Egyptian poultry slaughterhouses [82]. Moreover, in two Iraqi sheep slaughterhouses, Candida albicans was isolated by 10% in sheep organs (n = 10 isolates), by 8% in equipment swabbed before the slaughter process (n = 4 isolates), and by 12% in equipment swabbed after slaughter (n = 6 isolates) [11]. Candida albicans was also one of the fungal species that were previously described to be carried by flies (Musca domestica) present in an Iraqi slaughterhouse (n = 11 isolates) [43]. Candida albicans, a commensal organ-ism, is part of the microbiota in healthy individuals. However, under certain condi-tions, it can transition from a commensal to a pathogenic state [83]. Another recognized opportunistic pathogen is Candida tropicalis, which is consid-ered the most prevalent pathogenic yeast species within the Candida non-albicans group [84]. In humans, Candida tropicalis is associated with superficial mycoses, such as onychomycosis, otomycosis, oral and skin candidiasis, keratitis, and genital tract in-fections [85]. Candida tropicalis was reported in swabs from broiler carcasses (16.7%, n = 5 isolates) [82], from sheep organs (1%, n = 1 isolate) [11], and from flies (4.5%, n = 7 isolates) [43].

How do the findings regarding Candida spp. in this study fit into the broader understanding of the role of fungi in meat safety and occupational health in slaughterhouses? (lines 486 - 508)

Our reply: The findings on Candida spp. in these studies contribute to a broader understanding of the role of fungi in meat safety and occupational health in slaughterhouses, demonstrating that these microorganisms can be present on various surfaces, including carcasses, equipment, and biological vectors such as flies. Their presence may indicate inadequate hygienic and sanitary conditions, as well as potential meat spoilage. Moreover, vectors such as insects can contribute to the dissemination of Candida in the slaughterhouse context.

In terms of meat safety, the detection of Candida albicans and Candida tropicalis in carcasses and organs of ruminants suggests the possibility of contamination along the production chain. Although Candida spp. are not traditionally considered foodborne pathogens, there is evidence that Candida may be associated with gastroenteritis in immunocompromised patients.  In the context of occupational health, the presence of Candida spp. on equipment and in samples from workers raises concerns about occupational exposure. Because Candida spp. can act as opportunistic pathogens, immunocompromised workers may be at an increased risk of infection. Furthermore, it is clear that, like other fungi, the study of the role of Candida spp. and its impact on meat safety and occupational health in the context of slaughterhouses is still minimal. Thus, the findings reinforce the need for more research.

We also added the following statement at the end of the first paragraph.

Opportunistic Candida spp. infections, including foodborne illness [x – Sharma et al., 2017] pose a significant threat to immunocompromised individuals [x – Hemaid et al., 2021]. Candida albicans and Candida parapsilosis can cause invasive candidiasis in humans, and some of their strains can be transmitted through contaminated food [x – Mohamed et al., 2023]. Also, their biofilm formation capabilities in food processing facilities may contribute to the recurring contamination of meat products [x – Mohamed et al., 2023].

Added references:

Sharma, K., Chattopadhyay, U., & Naskar, K. (2017). Prevalence of Candida albicans in raw chicken and mutton meat samples sold in the open markets of Kolkata city of West Bengal. Int. J. Livest. Res, 7, 243-249.

Mohamed, H. M., Aljasir, S. F., Moftah, R. F., & Younis, W. (2023). Mycological evaluation of frozen meat with special reference to yeasts. Veterinary World, 16(3), 571.

Hemaid, A. S. S., Abdelghany, M. M. E., & Abdelghany, T. M. (2021). Isolation and identification of Candida spp. from immunocompromised patients. Bulletin of the National Research Centre, 45, 1-8.

  1. Reviewer´s comment:

Lines 533–551: Human fungal infections caused by Rhodotorula spp. have been increasing over the last few decades, and these are considered emerging pathogens that primarily affect immunocompromised individuals [91]. Reports indicate that Rhodotorula spp. can cause fungaemia, meningitis, cutaneous infections, peritonitis, keratitis, ventriculitis, and other less common conditions [91]. Rhodotorula spp. were previously isolated from air samples in two Italian poultry slaughterhouses [45]. Rhodoturula spp. was also recovered from broiler carcass swabs (26.7%, n = 8 isolates) in an Egyptian study and represented 6% of the detectable genera isolated in a study carried out in Korean swine slaughterhouses [5]. Rhodotorula mucilaginosa (formerly Rhodotorula rubra) was detected in swabs from equipment taken before (18%, n = 9 isolates) and after slaughter (12%, n = 6 isolates) [11]. This strain was also recovered from air samples collected in a cutting site [45] and blade swabs [38]. Rhodotorula mucilaginosa, although rarely seen as an opportunistic pathogen, is one of the most common causative species of fungemia and may be associated with endocarditis in patients with chronic kidney disease [91, 92]. Other Rhodotorula isolates include Rhodotorula aurantiaca and Rhodotorula minuta, both of which were isolated from air samples collected at an Italian poultry slaughterhouse [45].

What are the specific environmental factors in the slaughterhouses that might be contributing to the presence and isolation of Rhodotorula spp.?

Our reply: Although the optimal growth temperature for Rhodotorula spp. is between 28 and 22°C, Rhodoturula spp. are ubiquitous and can grow abundantly in extreme environments. For example, Rhodotorula mucilaginosa is present in different habitats and substrates, including cold and extreme environments. Since there are no studies that precisely determine the general conditions of relative humidity, water activity and temperature in slaughterhouses, it isn't easy to pinpoint the environmental conditions that favour the presence of Rhodotorula. However, since it is a ubiquitous species that can withstand extreme environments, the species can adapt to different temperatures within the slaughterhouse, ranging from the hottest areas of the slaughter line to the refrigeration areas.

We added the following statement:

Although there are no studies specifically focusing on the environmental conditions in slaughterhouses that favour the growth of Rhodotorula spp., this species is ubiquitous and capable of withstanding extreme environments [x – de Menezes et al., 2019]. Therefore, this species may be adapted to the varying temperatures found in slaughterhouses.

Added references:

de Menezes, G. C., Amorim, S. S., Gonçalves, V. N., Godinho, V. M., Simões, J. C., Rosa, C. A., & Rosa, L. H. (2019). Diversity, distribution, and ecology of fungi in the seasonal snow of Antarctica. Microorganisms, 7(10), 445.

  1. Reviewer´s comment:

Lines 543 – 545: Rhodotorula mucilaginosa (formerly Rhodotorula rubra) was detected in swabs from equipment taken before (18%, n = 9 isolates) and after slaughter (12%, n = 6 isolates) [11]. This strain was also recovered from air samples collected in a cutting site [45] and blade swabs [38].

How does the presence of Rhodotorula mucilaginosa on equipment before and after slaughter affect the risk of contamination to the meat?

Our reply: According to the study, no significant differences were observed in the proportions of yeast isolates before and after the slaughter process. However, the fact that this species is associated with biofilm formation, albeit with moderate capacity, and was detected at both sampling points suggests that biofilms may contribute to its resistance and persistence on equipment. This could indicate that the cleaning and disinfection procedures are insufficient to eliminate the organism or that these techniques are not being appropriately applied. Consequently, the presence of Rhodotorula mucilaginosa on contaminated equipment increases the risk of cross-contamination, potentially compromising meat safety.

We added the following statement:

According to the study [x - Mashari & Al-Haddad, 2024], no significant differences were observed in the isolation proportions of yeasts before and after the slaughter process. However, the fact that Rhodotorula spp. is associated with biofilm formation [x – Gattlen et al., 2011], including Rhodotorula mucilaginosa [x – Gharaghani et al., 2020], and was detected at both sampling points suggests that biofilms may contribute to its resistance and persistence on equipment. This could indicate that the cleaning and disinfection procedures are insufficient to eliminate the organism or that these techniques are not being correctly applied.

Added references:

Gharaghani, M., Taghipour, S., & Zarei Mahmoudabadi, A. (2020). Molecular identification, biofilm formation and antifungal susceptibility of Rhodotorula spp. Molecular Biology Reports, 47(11), 8903-8909.

Gattlen, J., Zinn, M., Guimond, S., Körner, E., Amberg, C., & Mauclaire, L. (2011). Biofilm formation by the yeast Rhodotorula mucilaginosa: process, repeatability and cell attachment in a continuous biofilm reactor. Biofouling, 27(9), 979-991.

  1. Reviewer´s comment:

Lines 547 – 549: Rhodotorula mucilaginosa, although rarely seen as an opportunistic pathogen, is one of the most common causative species of fungemia and may be associated with endocarditis in chronic kidney disease patients [91, 92].

Given that Rhodotorula mucilaginosa is associated with endocarditis in chronic kidney disease patients, what are the implications for workers' health in the slaughterhouse?

Our reply: Rhodotorula mucilaginosa is considered a rare opportunistic pathogen; however, its association with fungemia and endocarditis in patients with chronic kidney disease suggests that immunocompromised individuals may be at an increased risk of infection. In the context of occupational health in slaughterhouses, the presence of this fungus may have implications for workers exposed to humid environments and contaminated aerosols, especially those with predisposing factors such as immunosuppression, diabetes, or other chronic conditions.

Although the risk to healthy workers is low, the detection of R. mucilaginosa in the slaughterhouse environment highlights the importance of strict hygiene measures and the appropriate use of personal protective equipment (PPE). Furthermore, awareness of opportunistic fungal infections is relevant for occupational health surveillance, especially among vulnerable workers.

We added the following statement:

Therefore, awareness of opportunistic fungal infections is relevant for occupational health surveillance, especially among vulnerable workers.

  1. Reviewer´s comment:

Lines 550 – 551: Other Rhodotorula isolates include Rhodotorula aurantiaca and Rhodotorula minuta, both isolated from air samples collected from an Italian poultry slaughterhouse [45].

How do the different Rhodotorula species isolated (e.g., Rhodotorula aurantiaca, Rhodotorula minuta) compare in terms of their pathogenicity and potential risks in the slaughterhouse setting?

Our reply: In general, Rhodotorula spp. are yeasts widely distributed in the environment and rarely cause infections in immunocompetent individuals. However, some species, such as R. mucilaginosa and R. minuta, are recognised as opportunistic pathogens and can cause fungemia, endocarditis, and other invasive infections, especially in immunosuppressed patients. On the other hand, the pathogenicity of R. aurantiaca is less well documented, and there are few reports of its involvement in human infections. In the context of slaughterhouses, the presence of these species in air samples suggests a possible dispersion of fungi in the work environment, which may represent an occupational risk for immunocompromised workers or those exposed to contaminated aerosols. Although the direct impact of these species on meat safety is uncertain and has not been studied or documented, their ability to colonise equipment and surfaces reinforces the need for effective hygiene and air quality control practices to minimise risks to both occupational health and food safety.

We added the following statement:

Rhodotorula minuta, along with Rhodotorula mucilaginosa, is recognised as an opportunistic pathogen [x – Neves et al., 2019], whereas the pathogenicity of Rhodotorula au-rantiaca remains less understood.

Added references:

Neves, R. P., de Carvalho, A. M. R., da Silva, C. M., & Cerqueira, D. P. (2019). Rhodotorula spp. In Pocket Guide to Mycological Diagnosis (pp. 63-68). CRC Press.

Microbiological monitoring in slaughterhouses: the overlooked impact of fungi

  1. Reviewer´s comment:

Lines 617 – 618: Mycotoxin exposure requires the presence of mould and favourable conditions for toxin production, with humidity being a key factor.

What are the specific humidity levels and other environmental conditions within the slaughterhouse that are most conducive to mycotoxin production by moulds?

Our reply: We believe that this question was previously partially addressed. As previously mentioned, no studies have reported specific conditions of relative humidity, temperature, and water activity in the various slaughterhouse locations. It is only mentioned that there are high levels of moisture. However, there is data on the ideal conditions for mycotoxin production. The study of the ecology of fungi in slaughterhouses, explicitly examining the environmental conditions within these facilities that contribute to the growth of fungi and the production of mycotoxins, is urgent.

We added the following statement:

However, there is a notable gap in the study of environmental factors in slaughterhouses that may contribute to mycotoxin production by moulds and how these conditions may influence meat safety and occupational health.  

  1. Reviewer´s comment:

Lines 620–622: Assessing occupational risks involves considering the concentration of airborne mycotoxins, exposure duration, and frequency. Two main scenarios exist: regular low-level exposure or occasional high peaks. Health risk evaluation remains challenging due to the absence of regulatory limits for airborne mycotoxins.

How can we more accurately assess occupational risks associated with mycotoxin exposure, particularly given the absence of regulatory limits for airborne mycotoxins?

Our reply: Accurately assessing occupational risks associated with airborne mycotoxin exposure, despite the lack of regulatory limits, requires a multifaceted approach. Continuous environmental monitoring through air sampling in different slaughterhouse areas could help detect and quantify mycotoxins. Biomonitoring workers, by analysing specific biomarkers in biological fluids (such as urine, blood, or saliva), provide direct insights into absorption and metabolism. Also, epidemiological studies assessing respiratory, inflammatory, and other adverse effects in exposed workers further contribute to risk evaluation. The guidelines of the WHO consider an airborne fungal spore concentration (AFSC) above 500 CFU/m³ hazardous for the occupant and above 1000 CFU/m³ highly hazardous.

We added the following statement:

According to Viegas et al. [x – Susana, 2020], the key aspects that should be considered when occupational exposure assessment to mycotoxins is planned or performed may include the recognition of the possible presence of mycotoxins in the workplace, identifying possible exposure/contamination sources, collect contextual information, characterise exposure variability, and define Similar Exposure Groups (SEGs).

Airborne fungi are often used as indirect indicators of mycotoxins [x – Viegas et al., 2018]. However, in addition to the unreliability of this approach, as mycotoxins can persist even after fungi are eliminated [x – Viegas et al., 2018], there is also a lack of standardisation in fungal identification in slaughterhouses—alternative methods for assessing mycotoxin exposure also present limitations. PCR may overestimate or underestimate contamination, and ELISA-based biomonitoring cannot distinguish between occupational and dietary exposure [x - Marcelloni et al., 2024]. Thus, a more accurate assessment of occupational risks related to airborne mycotoxin exposure requires an integrated approach that combines various sampling and laboratory methods, including culture-based, PCR, High-Performance Liquid Chromatography, ELISA, and cytotoxicity tests [Marcelloni et al., 2024].

The lack of epidemiological studies makes it difficult to assess the acute and chronic health effects of mycotoxin exposure and to establish regulatory occupational exposure limits for individual mycotoxins and their mixtures [x – Viegas et al., 2018]. Given the World Health Organisation (WHO) guidelines on airborne fungal spore concentrations, correlating spore counts with mycotoxin presence could help establish risk thresholds. Considering these guidelines, airborne fungal spore concentrations above 500 CFU/m3 are considered hazardous, and above 1000 CFU/m3 may be highly hazardous [x - Al Hallak et al., 2023].

Added references:

Marcelloni, A. M., Pigini, D., Chiominto, A., Gioffrè, A., & Paba, E. (2024). Exposure to airborne mycotoxins: The riskiest working environments and tasks. Annals of Work Exposures and Health, 68(1), 19-35.

Al Hallak, M., Verdier, T., Bertron, A., Roques, C., & Bailly, J. D. (2023). Fungal contamination of building materials and the aerosolisation of particles and toxins in indoor air and their associated risks to health: a review. Toxins, 15(3), 175.

Viegas, S., Viegas, C., Martins, C., & Assunção, R. (2020). Occupational exposure to mycotoxins—different sampling strategies telling a familiar story regarding occupational studies performed in Portugal (2012–2020). Toxins, 12(8), 513.

  1. Reviewer´s comment:

Lines 623–624: The lack of monitoring for these compounds in workplaces makes it challenging to compare exposure levels, highlighting the need for standardised methods.

What would be the most effective standardised method for monitoring mycotoxins in the workplace, given the current challenges?

Our reply: We believe that this question has already been addressed. The most common sampling methods for mycotoxin exposure include settled dust collection, ambient air sampling using impingers and impactors on nutrient media (MEA and DG18), and personal air pumps with inhalable fraction selectors (IOM or CIP 10) for task-based exposure assessment. High-efficiency dry filter air samplers (SASS 3100) have proven unsuitable due to their inability to retain small particles carrying mycotoxins. Combining active and passive sampling is increasingly recommended, as active methods capture short-term contamination (minutes), while passive methods reflect long-term exposure (days to months). High-performance liquid chromatography-tandem mass spectrometry (HPLC-MS/MS) is the preferred analytical technique for detecting multiple toxins with high sensitivity and selectivity. PCR is used to identify mycotoxin-producing moulds but is becoming less common. ELISA tests are available for detecting mycotoxins in human biological fluids, supporting biomonitoring. However, biomonitoring alone cannot distinguish whether exposure is due to food intake or workplace contamination, requiring a control group for an accurate assessment. Many researchers advocate for a multi-approach strategy, combining different sampling and laboratory methods (culture-based, PCR, HPLC, ELISA, cytotoxicity tests) to improve risk characterisation.

  1. Reviewer´s comment:

Lines 625 – 627: Airborne fungi are often used as indirect indicators of mycotoxins [99]. However, in addition to the unreliability of this approach, as mycotoxins can persist even after fungi are eliminated [99], there is also a lack of standardisation in fungal identification in slaughterhouses.

How reliable are the alternative methods to using airborne fungi as indirect indicators of mycotoxins, considering the persistence of mycotoxins after fungi elimination?

Our reply: Alternative methods to using airborne fungi as indirect indicators of mycotoxins have limitations that affect their reliability. PCR-based detection of toxigenic species does not necessarily predict mycotoxin presence, as it may overestimate contamination due to non-mycotoxin-producing fungi or underestimate it since mycotoxins can persist after fungal death. ELISA assays enable the detection of mycotoxins in human biological fluids, such as urine and blood; however, biomonitoring alone cannot differentiate between occupational and dietary exposure. To improve accuracy, 24-hour urine samples or first-morning voids are recommended over spot samples, as variations in urinary excretion are more pronounced for toxins like DON. HPLC-MS/MS is the most reliable method for detecting multiple mycotoxins in a single step, offering high sensitivity and selectivity. Given these challenges, many authors recommend a multi-approach strategy, combining active and passive environmental sampling, biomonitoring, and analytical techniques (such as PCR, HPLC, ELISA, and cytotoxicity tests) to obtain a more comprehensive and accurate risk assessment.

We believe that the question has already been addressed.

  1. Reviewer´s comment:

Lines 634 – 635: Protective measures should be taken to reduce the potential exposure to fungal burden. Preventive measures should be implemented, such as disinfection and the use of personal protective devices, including filtration masks and gloves. Increased awareness and education on this topic should be provided to both workers and employers.

How can we effectively increase awareness and education on mycotoxin exposure risks among workers and employers in the slaughterhouse setting? 

Our response: Raising awareness about mycotoxin exposure in slaughterhouses requires practical, engaging training that is tailored to both workers and employers. Demonstrations conducted in small groups, using the workers' native language, ensure better understanding. Pictorial posters and strategically placed visual aids reinforce key messages. Employers should integrate safety measures into health protocols, collaborate with regulatory bodies, and establish regular training schedules. Avoiding written instructions and using interactive methods can enhance participation, fostering a safer work environment.

We added the following statement:

Increased awareness and education on this topic should be provided to both workers and employers. During worker training sessions, topics covered may include good hygiene practices, cleaning and disinfection protocols and, among other potential occupational health risks, those associated with occupational exposure to mycotoxins. Also, explaining and increasing awareness regarding possible meat contamination sources should be beneficial for meat safety in the slaughterhouse context. Mother-shaw et al. [x, 2006] previously suggested that demonstrations conducted in small groups, in the workers' native language, ensure better understanding among slaughterhouse workers undergoing training. Avoiding written instructions and using interactive methods and strategically placed pictorial posters can enhance participation and aid in reinforcing key messages [x – Mothershaw et al., 2006].  

Added references:

Mothershaw, A. S., Consolacion, F., Kadim, I. T., & Al Raisi, A. N. (2006). The role of education and training levels of slaughterhouse workers in the cross-contamination of carcasses. International Journal of Postharvest Technology and Innovation, 1(2), 142-154.

  1. Reviewer´s comment:

Lines 617 – 637: Mycotoxin exposure requires the presence of mould and favourable conditions for toxin production, with humidity being a key factor [98]. However, exposure can occur even without visible mould, as small fungal biomass may produce significant mycotoxins. Assessing occupational risks involves considering the concentration of airborne mycotoxins, exposure duration, and frequency. Two main scenarios exist: regular low-level exposure or occasional high peaks. Health risk evaluation remains challenging due to the absence of regulatory limits for airborne mycotoxins [98]. The lack of monitoring of these compounds in workplaces makes it difficult to compare exposure levels, highlighting the need for standardised methods. Airborne fungi are often used as indirect indicators of mycotoxins [99]. However, in addition to the unreliability of this approach, as mycotoxins can persist even after fungi are eliminated [99], there is also a lack of standardisation in fungal identification in slaughterhouses. Additionally, insufficient epidemiological studies hinder the assessment of acute and chronic health effects of mycotoxin exposure, making it crucial to establish occupational exposure limits for individual mycotoxins and their mixtures [99]. Protective measures should be taken to reduce the potential exposure to fungal burden. Preventive measures should be implemented, such as disinfection and the use of personal protective devices, including filtration masks and gloves [13]. Increased awareness and education on this topic should be provided to both workers and employers. Ultimately, it is essential to emphasise that the presence of fungi capable of producing mycotoxins in slaughterhouse wastewater poses an environmental risk [100].

How do the risks associated with mycotoxin exposure in the slaughterhouse compare to those in other food production or industrial settings?

Our reply:

We added the following statement:

Although the available studies do not allow for a reliable comparison, slaughter-house workers may present a lower risk in comparison to other food industries and industrial settings, such as grain processing and feed production, that tend to frequent and direct handle highly contaminated raw materials [x – Viegas (Susana) et al., 2020]. As mentioned, Viegas et al. [x – Susana, 2015] confirmed aflatoxin B1 occupational exposure levels in a poultry slaughterhouse, although these levels were lower than those found in poultry production. The authors [x – Susana 2015] explain that this may be because, since the chickens are slaughtered at the beginning of the process and, after that, workers in most workplaces wear gloves when handling the birds, not all environments in slaughterhouses present conditions conducive to exposure. Furthermore, the strict hygienic conditions applied in this environment can help eliminate and prevent the contamination and spread of fungi, something impossible to guarantee in poultry production.

Added references:

Viegas, S., Viegas, C., Martins, C., & Assunção, R. (2020). Occupational exposure to mycotoxins—different sampling strategies telling a familiar story regarding occupational studies performed in Portugal (2012–2020). Toxins, 12(8), 513.

Viegas, S., Veiga, L., Almeida, A., dos Santos, M., Carolino, E., & Viegas, C. (2016). Occupational exposure to aflatoxin B1 in a Portuguese poultry slaughterhouse. Annals of Occupational Hygiene, 60(2), 176-183.

Conclusion

  1. Reviewer´s comment:

Lines 646 – 648: Further research would be essential to establish the limits of mycotoxins in meat and occupational exposure limits and assess the long-term health impacts of mycotoxin exposure in the slaughterhouse. Furthermore, implementing standardised fungal monitoring programs, integrating mould control and cleaning into SSOPs, and raising awareness among workers, employers, and regulatory entities about the issue are essential steps to enhance public health.

How can standardised fungal monitoring programs be effectively designed and implemented in the unique context of slaughterhouses? (lines 646 - 648) What are the most appropriate methods for establishing the limits of mycotoxins in meat and occupational exposure limits?

Our reply: Regular and planned air and surface sampling, integration of mould control into Standard Sanitation Operating Procedures (SSOPs), worker training, and the use of biomarkers to monitor occupational exposure are essential for implementing standardised fungal monitoring programs in slaughterhouses. Establishing mycotoxin limits in meat and occupational exposure should be based on toxicological studies, sensitive analytical techniques, and epidemiological assessments to define regulatory standards, ensuring food safety and protecting worker health.

The outlined lines were corrected to:

Furthermore, implementing regular and planned standardised fungal monitoring programs in air and surface samples, integrating mould control and cleaning into Standard Sanitary Operating Procedures (SSOPs), and raising awareness among workers, employers, and regulatory entities about the issue are essential steps to enhance public health. Establishing mycotoxin limits in meat and addressing occupational exposure should be a priority. This should be based on toxicological studies, sensitive analytical techniques, and epidemiological assessments to define regulatory standards, thereby ensuring food safety and protecting worker health.
